# MALIBO: META-LEARNING FOR LIKELIHOOD-FREE BAYESIAN OPTIMIZATION

## ABSTRACT

Bayesian Optimization (BO) is a popular method to optimize expensive black-box functions. While BO typically only optimizes a single task, recent methods exploit knowledge from related tasks to warm-start BO and improve data-efficiency. However, these methods are either not scalable or sensitive to heterogeneous value scales across multiple tasks. We propose a novel approach to solve these problems by combining meta-learning with a likelihood-free acquisition function. Specifically, our meta-learning model simultaneously learns the underlying (task-agnostic) data distribution and a latent feature representation for individual tasks to be used as the acquisition function inside BO. The likelihood-free approach has less stringent assumptions about the problems compared to regression based methods and works with any classification algorithm, making it computation efficient and robust to different scales across tasks. Finally, we use gradient boosting as a residual model on top to adapt to distribution drifts between new and prior tasks, which might otherwise weaken the usefulness of the meta-learned features. Experiments show that the meta-model learns an effective prior for warm-starting optimization algorithms, is cheap to evaluate, and invariant under changes of scale across different datasets.

## 1 INTRODUCTION

Bayesian Optimization (BO) is a widely used method to optimize expensive black-box functions (Shahriari et al., 2016) and has been successfully applied in different fields, including automated machine learning (ML) (Hutter et al., 2019). Given small amounts of data, traditional BO uses a Gaussian Process (GP) surrogate model together with an acquisition function to quickly optimize a black-box function. However, most BO techniques start from scratch for each new optimization problem, instead of leveraging information from previous runs for similar tasks to further improve data-efficiency.

To warm-start BO, exploiting additional task information has been explored in the context of transfer learning (Weiss et al., 2016) and meta-learning (Vanschoren, 2018). Prior knowledge can be used to build informed surrogate models (Schilling et al., 2016; Wistuba et al., 2018; Feurer et al., 2018b; Perrone et al., 2018), to restrict the search space (Perrone et al., 2019), or to warm-start the optimization with configurations that generally score well (Feurer et al., 2014; Salinas et al., 2020).

However, these approaches have three important issues:

(i) GPs scale poorly due to their cubical computational complexity (Rasmussen, 2004).

(ii) The standard BO framework requires a surrogate model with well-calibrated and tractable predictive uncertainty, which is challenging in high-dimensional problems (Tiao et al., 2021; Song et al., 2022).

(iii) Regression models, including GPs, struggle with different scales and noise levels across tasks, which hurts warm-starting and optimization efficiency (Feurer et al., 2018a).

We propose a new meta-learning BO approach that can effectively transfer knowledge from related tasks and scales to large datasets. Our method is inspired by the idea of likelihood-free BO (Bergstra et al., 2011; Tiao et al., 2021; Song et al., 2022), which replaces the surrogate model with a meta-learned classifier that directly balances exploration and exploitation without modeling the

objective function. That way, we elegantly avoid both the scalability and the scale sensitivity issues simultaneously. We make the following contributions:

(i) A novel probabilistic meta-learning model that uses Bayesian logistic regression and a probabilistic approach to learn feature representations from prior tasks.

(ii) A scalable BO technique with good anytime performance that combines a meta-learning classifier and a likelihood-free acquisition function.

(iii) Robust adaptation to new tasks by combining the meta-learned classifier with gradient boosting to correct prediction errors and Thompson Sampling for more explorationn.

## 2 RELATED WORK

**Meta-learning**     Various methods have been proposed that improve the data-efficiency of Bayesian optimization (BO) by leveraging the information of previous observations from similar tasks. They apply meta-learning (Vanschoren, 2018) or transfer-learning (Weiss et al., 2016) depending on the context and have been proven effective in various applications (Andrychowicz et al., 2016; Finn et al., 2017). We refer to Vanschoren (2018) for an in-depth overview.

One line of work adapts the initial design to warm-start BO, either by reducing the search space (Perrone et al., 2019; Li et al., 2022) or reusing good configurations from similar tasks, where similarity can be based on hand crafted features (Feurer et al., 2014) or learned with Neural Networks (NNs) (Kim et al., 2017). Alternative approaches estimate the usefulness of a given configuration on both the current and prior tasks based on heuristics (Wistuba et al., 2015) or learning-based (Volpp et al., 2020) methods. Other approaches use transfer-learning to modify the probabilistic surrogate model, for instance using a multi-task GP (Swersky et al., 2013; Tighineanu et al., 2022), an additive GP model (Golovin et al., 2017; Marco et al., 2017), or weighted combinations of independent GPs for different tasks (Schilling et al., 2016; Wistuba et al., 2018; Feurer et al., 2018a).

Several methods *simultaneously* learn the initial design and modify the surrogate model. Springenberg et al. (2016) apply task-specific embeddings for BO and use a Bayesian NN as the surrogate model, which are computationally expensive and hard to train. Perrone et al. (2018) propose Adaptive Bayesian Linear Regression (ABLR), which uses a NN to learn a shared feature representation across tasks with task-specific BLR layers to improve scalability and adaptability. However, both these methods are sensitive to changes in the scale and noise level across datasets. To tackle this, Salinas et al. (2020) propose Gaussian Copula Process Plus Prior (GC3P), which transforms the data response values via the empirical CDF, and fits a NN across all prior tasks. This NN is used to warm-start the optimization and predict the mean for a GP on the target task. Despite its robustness, the use of a GP surrogate still limits its applicability on high-dimensional problems.

Our meta-learning method is closely related to *Bayesian optimization with NNs and embedding reasoning* (BANNER, Berkenkamp et al. (2021)), which uses a meta-learning model based on a NN to learn a latent representation and a task-specific BLR layer, similar to ABLR. However, the model output is divided into a task-independent mean and a task-specific residual prediction learned by a BLR layer. In this paper, we introduce a classifier variant of this meta-learning model and combine it with a likelihood-free acquisition function.

**Likelihood-free Bayesian Optimization**     Bayesian optimization does not require an explicit model of the likelihood of the observed values (Garnett, 2022). Tree-structured Parzen Estimators (TPE, Bergstra et al. (2011)) phrase BO as a density ratio estimation problem (Sugiyama et al., 2012) and use the density ratio over 'good' and 'bad' configurations as an acquisition function without a probabilistic regression model. Tiao et al. (2021) estimate the density ratio through class probability estimation (Qin, 1998), which is equivalent to modeling the acquisition function with a binary classifier. Likelihood-Free BO (LFBO, Song et al. (2022)) improves upon this by weighting the observations.

Likelihood-free BO approaches address two drawbacks of traditional GP-based BO methods: the computationally expensive inference and the lack of flexibility due to the strong assumptions of most kernel methods. Rather than modeling the objective function, likelihood-free BO methods can use deterministic classifiers to separate good and bad configurations resulting in scale-invariant models

and allows the application of any binary classification method (Tiao et al., 2021; Song et al., 2022). We leverage this flexibility to create a new meta-learning classifier, which yields a scalable method that is robust to heterogeneous scales across datasets.

## 3 PROBLEM STATEMENT AND BACKGROUND

In this section, we introduce our problem setting and introduce related methods.

### 3.1 BAYESIAN OPTIMIZATION

Bayesian optimization (BO) aims to optimize a black-box function $f(\boldsymbol{x}) : \mathcal{X} \to \mathbb{R}$ over $\boldsymbol{x} \in \mathcal{X}$. At each step $n$, BO proposes a $\boldsymbol{x}_n$ obtains a noisy observation $y_n = f(\boldsymbol{x}_n) + \epsilon_n$. The proposal is based on a probabilistic surrogate model $\mathcal{M}$ and all previous observations $\mathcal{D}_{n-1} = \{(\boldsymbol{x}_i, y_i)\}_{i=1}^{n-1}$. The models posterior prediction $p(y \mid \boldsymbol{x}, \mathcal{D}_{n-1})$ combined with the acquisition function $\alpha(\boldsymbol{x}; \mathcal{D}_{n-1})$ quantifies the utility of each input and $\boldsymbol{x}_n = \arg\max_{\boldsymbol{x} \in \mathcal{X}} \alpha(\boldsymbol{x}; \mathcal{D}_{n-1})$. Typically, BO algorithms use a Gaussian Process model and assume $\epsilon_n \sim \mathcal{N}(0, \sigma^2)$, for some unknown but fixed variance.

Most acquisition functions, including Knowledge-Gradient (Frazier et al., 2009), Predictive Entropy Search (Hernández-Lobato et al., 2014) and Max-value Entropy Search (Wang & Jegelka, 2017) are defined as an expected utility function $U(y; \tau)$ usually with a threshold $\tau$ that heuristically balances exploration and exploitation. For example, the prevalent Expected Improvement (EI, Močkus (1975)) acquisition function has $U(y; \tau) := \max(\tau - y, 0)$ while the Probability of Improvement (PI, Kushner (1964)) has $U(y; \tau) := \mathbb{1}(\tau - y > 0)$. The expected utility over the posterior belief from the surrogate model $p(y \mid \boldsymbol{x}, \mathcal{D}_n)$ is given by

$$\alpha^U(\boldsymbol{x}; \mathcal{D}_n, \tau) = \mathbb{E}_{y \sim p(y \mid \boldsymbol{x}, \mathcal{D}_n)}[U(y; \tau)] = \int U(y; \tau) p(y \mid \boldsymbol{x}, \mathcal{D}_n) dy, \tag{1}$$

where the threshold is usually chosen as the lowest observed function value, i.e., $\tau = \min_{D_n} y_i$.

### 3.2 LIKELIHOOD-FREE ACQUISITION FUNCTIONS

Likelihood-free acquisition functions model the belief of a candidate being promising instead of explicitly computing the likelihood of outcomes via the model's posterior $p(y \mid \boldsymbol{x}, \mathcal{D}_N)$. One of the first likelihood-free BO algorithms, called Tree-structured Parzen Estimators (TPE, Bergstra et al. (2011)), dismisses the surrogate for the outcomes and models the two densities $\ell(\boldsymbol{x}) = p(\boldsymbol{x} \mid y \leq \tau, \mathcal{D}_n)$ and $g(\boldsymbol{x}) = p(\boldsymbol{x} \mid y > \tau, \mathcal{D}_n)$ instead. The threshold $\tau$ relates to the $\gamma$-th quantile of the observed $y$ values via $\gamma = \Phi(\tau) := p(y \leq \tau \mid \mathcal{D}_n)$. The density ratio (DR) serves as the acquisition function $\alpha$: $\alpha^{\mathrm{DR}}(\boldsymbol{x}; \mathcal{D}_N, \tau) = \ell(\boldsymbol{x})/g(\boldsymbol{x})$.

Tiao et al. (2021) propose to improve several aspects of TPE by directly estimating the DR rather than solving the more challenging problem of modeling two independent densities as an intermediate step. Their approach, dubbed BORE, rephrases the DR estimation as a binary classification problem when using the training loss

$$\mathcal{L}^{\mathrm{BORE}}(\boldsymbol{\theta}; \mathcal{D}_N, \tau) = -\frac{1}{N} \sum_{n=1}^{N} (k_n \log C_{\boldsymbol{\theta}}(\boldsymbol{x}_n) + (1 - k_n) \log(1 - C_{\boldsymbol{\theta}}(\boldsymbol{x}_n))). \tag{2}$$

Here $k_n = \mathbb{1}(y_n \leq \tau)$ represents the binary class labels estimated by the classifier $C_{\boldsymbol{\theta}}$ with learnable parameters $\boldsymbol{\theta}$. Specifically, they show $\alpha^{\mathrm{DR}}(\boldsymbol{x}; \mathcal{D}_N, \tau) \propto C_{\boldsymbol{\theta}}(\boldsymbol{x})$.

Although Tiao et al. (2021) argue that BORE resembles EI, assigning the same label to all observations with $y < \tau$ regardless of the magnitude of improvement conforms to the definition of PI rather than EI, and might lead to conservative optimization with little global exploration (Garnett, 2022; Song et al., 2022). In practice, EI has exhibits stronger and robuster performance than PI. Song et al. (2022) convert the expected utility of EI into an optimization problem using a variational representation and reformulate it as a classification problem with the following objective:

$$\mathcal{L}^{\mathrm{LFBO}}(\boldsymbol{\theta}; \mathcal{D}_N, \tau) = -\mathbb{E}_{(\boldsymbol{x}, y) \sim p(\boldsymbol{x}, y \mid \mathcal{D}_N)}[\max(\tau - y, 0) \log C_{\boldsymbol{\theta}}(\boldsymbol{x}) + \log(1 - C_{\boldsymbol{\theta}}(\boldsymbol{x}))], \tag{3}$$

The resulting method, dubbed Likelihood-Free Bayesian Optimization (LFBO), can be seen as a weighted classification problem with noisy targets for only class $k = 1$, where the EI utility function

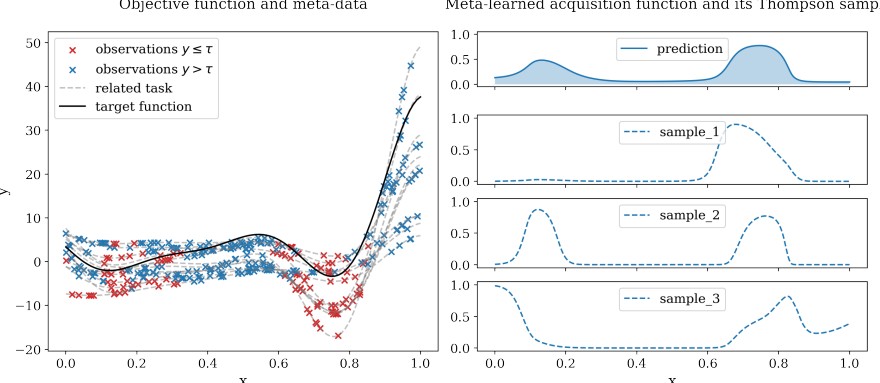

Figure 1: Illustration of meta-learning the acquisition function. Left: Observations (crosses) from 10 related tasks and the target task. The top performing observations ($\gamma = 1/3$) in each task are shown in red, the rest in blue. Right: The top panel shows the approximated predictive distribution (see Eq. (7)) while the others show Thompson samples. MALIBO successfully identifies the promising areas, and the Thompson samples show variability in the meta-learned acquisition function.

$\max(\tau - y, 0)$ weights the importance for observations below $\tau$ by their improvement. The minimizer of Eq. (3) is shown to be equivalent to the EI acquisition function (Song et al., 2022).

## 4 METHODOLOGY

In this section, we introduce our MetA-learning for LIkelihood-free BO method, dubbed MALIBO. It incorporates a probabilistic meta-learning approach derived from BANNER into the LFBO framework. In particular, we show how to convert BANNER's regression model into a meta-learning classifier that warm-starts BO and combine it with gradient boosting (Friedman, 2001). This yields good anytime performance, more flexibility in selecting the model, and better adaptation to new tasks. We first explain the different components and then summarize our algorithm in Algorithm 1.

### 4.1 META-LEARNING

Meta-learning for optimization strives to extract information from previous tasks and use this to accelerate the optimization of a new one. As all methods discussed in Section 2, we only consider the case of identical input spaces $\mathcal{X}$ across all tasks, which simplifies the learning problem. Our goal is to learn a probabilistic model for the likelihood-free acquisition function from the meta-data, as illustrated in Fig. 1 for a one-dimensional function.

Following the approach from Berkenkamp et al. (2021) and Perrone et al. (2018), our meta-learning classifier uses a deterministic, task-agnostic model to map the input space into a feature space, $h(\boldsymbol{x}) = \boldsymbol{\phi}$ of a predefined dimensionality. In this feature space, we seek to find a simple model able to classify the data on all prior tasks using a task specific adaptation. We propose to use Bayesian logistic regression with a minor modification. In addition to the standard linear model $\boldsymbol{\phi} \cdot \boldsymbol{z}_t$, where $\boldsymbol{z}_t$ represents prior task $t$ in the learned feature space, and the sigmoid function $\sigma$ converting the linear model into class probabilities, we introduce a task agnostic mean function $m(\boldsymbol{\phi})$ that acts as a non-linear bias. We provide an overview of our method in Fig. 2.

The meta-learning model $g(\cdot) = m(h(\boldsymbol{x}))$ learns a global prior for the classification problem, and the task-specific embedding vectors $\boldsymbol{z}_t$ representing the necessary adaptations. For better inference on new tasks, we regularize the distribution of the $\boldsymbol{z}$ values during training. We follow the technique used by Berkenkamp et al. (2021) and Saseendran et al. (2021) to bring $\boldsymbol{z}_t$ close to the prior distribution $p(\mathcal{Z}) = \mathcal{N}(\boldsymbol{0}, \mathbf{I})$, using a modified Kolmogorov-Smirnov test and the covariance to calculate the

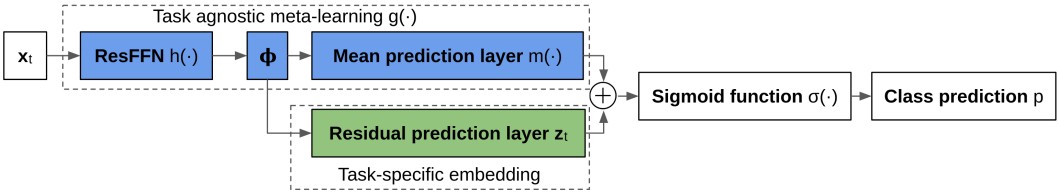

Figure 2: Schematic representation of our meta-learning classifier. A Residual Feedfoward Network (ResFFN) maps the input $\boldsymbol{x}_t$ to a latent feature representation $\boldsymbol{\phi}$. From this, a global mean prediction $m(\boldsymbol{\phi})$ and task-specific embedding $\boldsymbol{z}_t$ are combined and translated into a class prediction via a sigmoid function.

disparity of two distributions. The loss used for training on the meta-data reads:

$$\mathcal{L}^{\mathrm{meta}} = \min_{g(\cdot),\boldsymbol{z}_1,\dots,\boldsymbol{z}_T} \frac{1}{T} \sum_{t=1}^{T} \mathcal{L}^{\mathrm{LFBO}}(k_t; g(\boldsymbol{x}_t), \boldsymbol{z}_t) + \lambda \mathcal{R}(\{\boldsymbol{z}_1,\dots,\boldsymbol{z}_T\}; p(\mathcal{Z})), \quad (4)$$

where we average the contribution of all $T$ prior-tasks and $\mathcal{R}$ is the regularization term outlined by Berkenkamp et al. (2021) weighted by $\lambda$.

### 4.2 Task adaptation via Bayesian logistic regression

After training the meta-model, we need to adapt our predictions to each new task. Hence, we must estimate a value for $\boldsymbol{z}$ that yields good classification results[1] for the predictions $C(\boldsymbol{x}) = \sigma(g(\boldsymbol{x}) + \boldsymbol{z} \cdot h(\boldsymbol{x}))$. While we could greedily optimize Eq. (3) , we pursue a Bayesian approach, Bayesian logistic regression in particular, to capture the uncertainty in the task-specific $\boldsymbol{z}$. Especially with few observations on the new task, using a single $\boldsymbol{z}$ vector can introduce a strong bias.

Besides stronger exploration in the early phases of optimization, the Bayesian approach allows us to employ Thompson sampling (Thompson, 1933) for $\boldsymbol{z}$, as shown in Fig. 1. We believe this to be a valuable strategy for parallelization briefly explored in Appendix E. Kandasamy et al. (2018) showed that this bypasses the sequential scheme of traditional BO, without introducing the common computational burden of more sophisticated methods. See Garnett (2022) for more details.

Bayesian logistic regression is a simple yet powerful classification model with Bayesian treatment. Although exact Bayesian inference is intractable, the Laplacian and probit approximation allow us to approximate the weights' posterior and the predictive distribution respectively while remaining reliable and scalable (Bishop & Nasrabadi, 2006; Murphy, 2012). The Laplacian method fits a Gaussian distribution around the maximum-a-posteriori (MAP) estimate of the weights distribution and matches the second order derivative at the optimum.

In the first step, we obtain the MAP estimate by maximizing the posterior of our classifier $C$ parameterized by $\boldsymbol{z}$. To be consistent with the regularization used during meta-training, we use a standard, isotropic Gaussian prior for the weights: $p(\boldsymbol{z}) = \mathcal{N}(\boldsymbol{z} \,|\, \mathbf{m}_0, \boldsymbol{\Sigma}_0)$, with mean $\mathbf{m}_0 = \mathbf{0}$ and covariance matrix $\boldsymbol{\Sigma}_0 = \mathbf{I}$. Given observations $\mathcal{D}_N$, the log posterior likelihood w.r.t. $\boldsymbol{z}$ is given by

$$\mathcal{L}^{\mathrm{MALIBO}} = \frac{1}{2}(\boldsymbol{z} - \mathbf{m}_0)^T \boldsymbol{\Sigma}_0^{-1}(\boldsymbol{z} - \mathbf{m}_0) + \sum_{n=1}^{N} \left(k_n(\tau - y) \log C(\boldsymbol{x}_n) + \log(1 - C(\boldsymbol{x}_n))\right), \quad (5)$$

and defines the MAP estimate of the weights via $\boldsymbol{z}_{\mathrm{MAP}} = \arg\min_{\boldsymbol{z} \in \mathcal{Z}} \mathcal{L}^{\mathrm{MALIBO}}$. As a second step for the Laplace approximation, we compute the negative Hessian of the log posterior

$$\boldsymbol{\Sigma}_N^{-1} = -\nabla\nabla \ln p(\boldsymbol{z} \,|\, \mathcal{D}_N) = \boldsymbol{\Sigma}_0^{-1} + \sum_{n=1}^{N} \hat{k}_n(1 - \hat{k}_n)\boldsymbol{\phi}_n \boldsymbol{\phi}_n^T \quad (6)$$

which serves as the precision matrix for the approximated posterior $q(\boldsymbol{z}) = \mathcal{N}(\boldsymbol{z} \,|\, \boldsymbol{z}_{\mathrm{MAP}}, \boldsymbol{\Sigma}_N)$. To finally compute the approximate predictive distribution, we need to marginalize w.r.t. $p(\boldsymbol{z} \,|\, \mathcal{D}_N)$:

$$C(\boldsymbol{x}; g(\boldsymbol{x}), \mathcal{D}_N) = \int p(k = 1 \,|\, g(\boldsymbol{x}), \boldsymbol{z})p(\boldsymbol{z} \,|\, \mathcal{D}_N)d\boldsymbol{z} \simeq \int \sigma(\boldsymbol{z}^T \boldsymbol{\phi} + m(\boldsymbol{\phi}))q(\boldsymbol{z})d\boldsymbol{z} \quad (7)$$

---

[1]In contrast to the meta-training, we only infer $\boldsymbol{z}$ but keep the task agnostic model $g(\cdot)$ fixed.

---

**Algorithm 1:** MALIBO: Meta-learning for likelihood-free Bayesian optimization

---

    **Meta-learning:**
        **Input:** Meta-datasets $\mathcal{D}_t^{\text{meta}}$ for tasks $t = 1, \ldots, T$, proportion $\gamma \in (0, 1)$
**1**      Generate binary class labels for meta-data using $\gamma$: $\mathbb{1}(y \leq \tau)$, where $\tau = \Phi^{-1}(\gamma)$
**2**      Learn the meta-learning model $C(\cdot; \mathcal{D}^{\text{meta}})$ by optimizing $\mathcal{L}^{\text{meta}}$ (Eq. (4))
    **Bayesian optimization with Meta-learning:**
        **Input:** Fixed meta-learned model $g(\cdot)$
**3**      Obtain the first best guess $\boldsymbol{x}_0$ from the meta-learned model $\mathcal{D} = \{(\boldsymbol{x}_0, f(\boldsymbol{x}_0) + \epsilon)\}$
**4**      For the gradient boosting variant, use $C(\cdot) \leftarrow GB(C(\cdot))$ as the classifier
**5**      **while** *Has budget* **do**
**6**            Estimate $\boldsymbol{z}_{\text{MAP}}$ by optimizing $\mathcal{L}^{\text{MALIBO}}$ (Eq. (5)) w.r.t. $\boldsymbol{z}$
**7**            Update precision matrix $\boldsymbol{\Sigma}_n^{-1}$ (Eq. (6))
**8**            **if** *Thompson sampling* **then**
**9**                Sample $\hat{\boldsymbol{z}} \sim \text{MVN}(\boldsymbol{z}_{\text{MAP}}, \boldsymbol{\Sigma})$
**10**               $\boldsymbol{x}^* = \arg\max_{\boldsymbol{x} \in \mathcal{X}} C(\boldsymbol{x}; \hat{\boldsymbol{z}}, g(\boldsymbol{x}), \mathcal{D})$
**11**            **else**
**12**               $\boldsymbol{x}^* = \arg\max_{\boldsymbol{x} \in \mathcal{X}} C(\boldsymbol{x}; g(\boldsymbol{x}), \mathcal{D})$ with probit approximation (Eq. (25))
**13**            $\mathcal{D} \leftarrow \mathcal{D} \cup \{(\boldsymbol{x}^*, f(\boldsymbol{x}^*) + \epsilon)\}$

---

Although no closed form solution for the integral in Eq. (7) is available due to the logistic activation, we can evaluate the predictions with probit approximation, which makes use of the similarity of the sigmoid function and the probit function. See Appendix B for further details.

### 4.3 GRADIENT BOOSTING AS A RESIDUAL PREDICTION MODEL

While the learned feature representation warm-starts the optimization, the model gains more knowledge about the new task with growing data and needs to adapt. Eventually, the accuracy of $C(\boldsymbol{x})$ will saturate, as the parametric model has only limited flexibility and is ultimately limited by the amound and the quality of the meta-data. We expect a suboptimal classification performance especially in cases of little meta-data or when a large discrepancy between the training data and the meta-data distribution exists[2]. More in depth studies are shown in Appendix D.2.

To counteract this, we propose to combine our method with gradient boosting (GB) (Friedman, 2001). In every iteration, after updating the Bayesian logistic regression model, we train a gradient boosting model (regression trees) on $\mathcal{D}_n$ without leveraging any meta-learned features and use our classifier as the first one in the ensemble. This way, GB only corrects where our meta-model is inaccurate and allows for fine-tuning and better convergence.

## 5 EXPERIMENTS

In this section, we describe the experiments conducted to empirically evaluate our method. For the choice of problems, we focus on automated machine learning (AutoML), i. e. hyperparameter optimization (HPO) and neural architecture search (NAS). Besides that, we evaluate our methods on synthetic functions to show the robustness against multiplicative noise. Throughout all benchmarks, we show that MALIBO clearly improves upon LFBO's performance.

We compare our method against state-of-the-art baselines across all problems. We picked random search (Bergstra & Bengio, 2012), the gradient boosting variants of BORE (Tiao et al., 2021) and LFBO (Song et al., 2022) and BO with GPs as baselines without meta-learning and chose ABLR (Perrone et al., 2018), RGPE (Feurer et al., 2018a) and GC3P (Salinas et al., 2020) as representative algorithms with a meta-learning component. Additionally, we consider the performance of our method without any additional components (MALIBO), with gradient boosting (MALIBO (GB)), with Thompson sampling (MALIBO (TS)), and with both (MALIBO (GB-TS)).

---

[2]A large missmatch between traning and test data can arise here when the $x_n$ all cluster around an optimum after many iterations, but the meta-data consisted of IID points on all tasks.

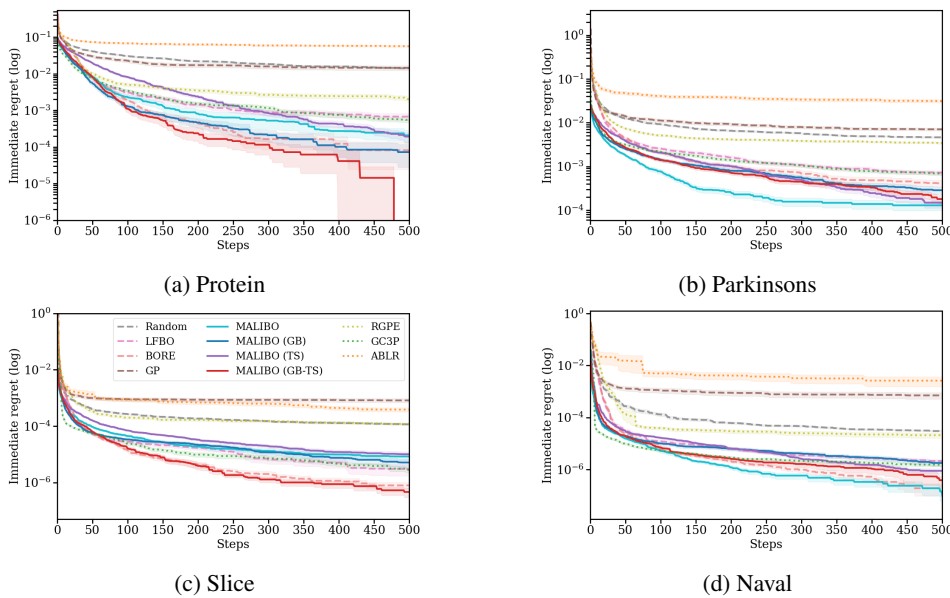

Figure 3: Immediate regret for different BO algorithms on the HPOBench neural network tuning problems ($D = 9$) for 4 datasets. The optimization objective in this benchmark is the validation loss.

We use *immediate regret* to quantitatively measure performance of different methods, which by definition is the absolute error between the global minimum and the lowest function value obtained so far. For all benchmarks, we report the results by mean and standard error across 100 random runs.

We implemented ABLR, RGPE and MALIBO in PyTorch (Paszke et al., 2019), and used scikit-learn (Pedregosa et al., 2011) for gradient boosting. As the meta-learning model for MALIBO, we considered a 4-layer, 64-unit residual feedforward network ResFFN-4-64. For more details we refer to Appendix G. For GC3P, we used the authors' open source implementation[3]. The algorithm samples five candidates from a meta-learned NN model before building a task-specific Copula process while BORE and LFBO sample 10 random configurations for gathering global information from the target task before training a model. Thanks to the meta-learned acquisition function, MALIBO starts the optimization at the point with highest acquisition function value. For all likelihood-free BO methods the required hyperparameter $\gamma$, we set $\gamma = 1/3$, following Tiao et al. (2021) and Song et al. (2022).

**Neural network tuning (HPOBench)** This benchmark represents a joint NAS and HPO for a two-layer feed-forward regression network on four popular UCI datasets (Dua & Graff, 2017). The search space is 9 dimensional and available as a tabular benchmark (Klein & Hutter, 2019; Eggensperger et al., 2021) with a total of 62,208 unique configurations. The optimization objective in this benchmark is the validation mean squared error after training with the corresponding network configuration (see Appendix H.1 for more details). As meta-data for each dataset, we randomly sampled 512 configurations from each of the remaining three. Fig. 3 shows the strong warm-starting performance of all MALIBO variants. For the protein and Slice dataset, MALIBO (TS-GB) exhibits the best final performance. For Parkinsons and Naval, the problem characteristics seem to favour the most basic MALIBO variant. GC3P performs very competitively often being the best method once the Copula process is fitted, but its final performance is usually matched or surpassed by BORE and LFBO. For all datasets, ABLR performs poorly, presumably due to the small number of prior-tasks and the fact that the meta-training seems to overfit to the meta-data in the first BO iterations. The GP based methods barely outperform RS, indicating that GPs fail to model the loss landscape.

**Neural architecture search (NASBench201)** NASBench201 (Dong & Yang, 2020) considers designing a neural cell with 6 discrete parameters totaling 15,625 unique architectures, evaluated on CIFAR-10, CIFAR-100 (Krizhevsky, 2009) and ImageNet-16 (Chrabaszcz et al., 2017). The

---

[3]https://github.com/geoalgo/A-Quantile-based-Approach-for-Hyperparameter-Transfer-Learning

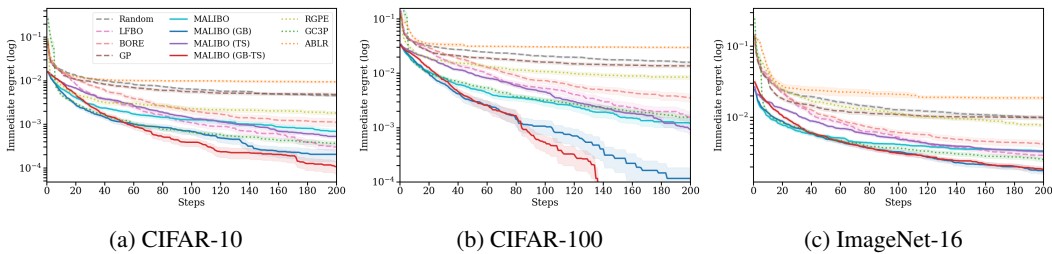

Figure 4: Immediate regret for different BO algorithms on the NASBench201 neural architecture search problems ($D = 6$) on 3 datasets. We optimize the validation accuracy for this benchmark.

optimization objective is validation accuracy after training the corresponding network architecture. We refer to Appendix H.3 for more details. We randomly sampled 512 configurations from each of two other datasets for meta-training. From the results shown in Fig. 4, we see that our methods start again with the lowest average regrets, and the gradient boosting variants converge the fastest compared to all baselines across all datasets. The graphs show that Thompson sampling (TS) varaints shows slightly slower convergence in the beginning, but they can outperform their non-TS counterparts. GC3P remains a strong competitor, but seems to converge prematurely. ABLR again suffers from few prior-tasks, and the other GP based methods struggle again with the loss surface.

**ML algorithms tuning (MLBench)** We picked 5 algorithms (SVM, LogReg, XGBoost, Random-Forest and MLP) from the ML benchmark suite provided by Eggensperger et al. (2021) evaluated on 20 OpenML tasks (Vanschoren et al., 2014), except for MLP with only 8 tasks. The search spaces dimensions range from 2 (SVM) to 5 (MLP) with the validation error as the objective. We refer to Appendix H.2 for more details. For meta-data, we randomly sampled 128 configurations from each task except the current target one (which changed across repetitions). As seen in Fig. 5, all MALIBO variants continue to demonstrate strong warm-starting performance and similar performance on all benchmarks, except for RandomForest, where gradient boosting variants are better. ABLR finally shows its potential with more meta-tasks and is competitive on some benchmarks, but fails to converge on most of them. The warm-starting of GC3P fluctuates from on-par with MALIBO (SVM, LogReg) to essentially random guesses (MLP, XGBoost). RGPE demonstrates quick adaptation, but the GP seems to limit convergences again.

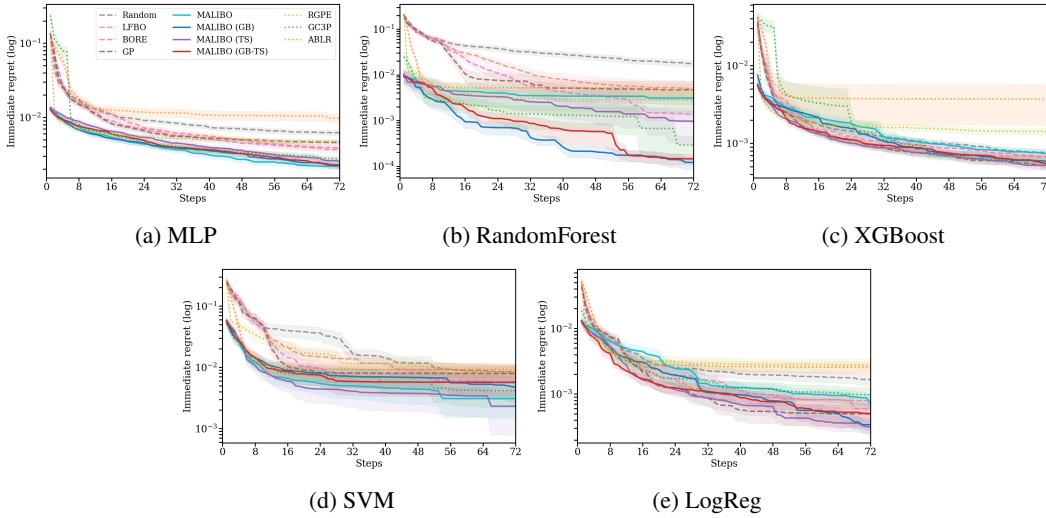

Figure 5: Immediate regret for different BO algorithms on the HPOBench hyperparamter tuning problems for 5 different machine learning algorithms. We specifically optimize the $(1 - accuracy)$ for this benchmark.

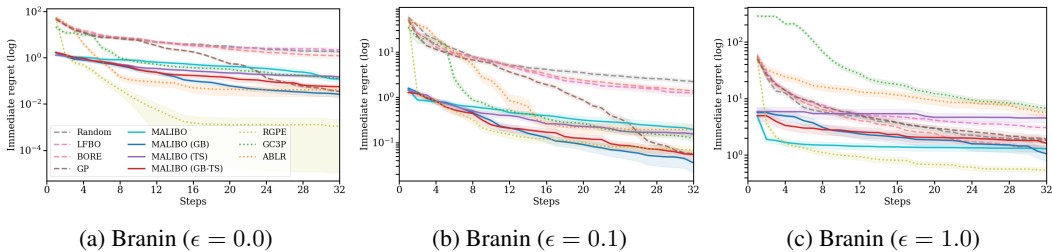

(a) Branin ($\epsilon = 0.0$)  (b) Branin ($\epsilon = 0.1$)  (c) Branin ($\epsilon = 1.0$)

Figure 6: Immediate regret for different BO algorithms on Branin function ensembles ($D = 2$) with different levels of multiplicative noise.

**Robustness against different noise level** To test the robustness of our method against different noise levels in the data, we use synthetic function ensembles introduced by Berkenkamp et al. (2021) with multiplicative noise. We focus on the Branin function ensemble (Dixon, 1978), which is a two-dimensional problem with 3 local minima. Their location and the global minimum varies across different functions in the ensemble. See Appendix H.6 for more details.

To avoid biasing this experiment towards a single method, we decided to use a heteroscedastic noise that is incompatible with any assumptions about the noise of any method. In particular, this violates the GP methods' and ABLR's assumption of homoscedastic, Gaussian noise. GC3P makes a similar assumption, but after the nonlinear transformation applied to the y-values, which does not translate to a well-known noise model. BORE, LFBO and by extension all MALIBO variants, make no explicit noise assumptions, but will optimize for the best mean. We choose a multiplicative noise, i.e. $y = f(x) \cdot (1 + \epsilon \cdot n)$, where $n \sim \mathcal{N}(0, 1)$. To test the robustness, we evaluate $\epsilon \in \{0, 0.1, 1.0\}$. The noise corrupts observations with large values more, while having a smaller effect on lower function values. For meta-training, we randomly sampled 128 noisy observations from 256 functions in the ensemble. We show our results in Fig. 6, where we can see across all magnitudes of multiplicative noise, our method still learns a meaningful prior for the optimization. As the noise level increases, the performaces of all methods degrade, with ABLR and GC3P performing worst for the largest noise. The GP based methods, especially RGPE do well on such smooth functions and handle the noise surprisingly well. In comparison, BORE, LFBO and MALIBO show the least performance losses with growing noise on a function ensemble where GP methods excel.

## 6 CONCLUSION

We introduced Meta-learning for Likelihood-free BO (MALIBO), a novel meta-learning optimization algorithm that is computationally efficient and robust to varying scales of the observations and heterogeneous noise. By directly modeling the acquisition function from observations, the method makes fewer assumptions about the data and noise distributions. Coupled with meta-learning, it leverages information from prior tasks for more sample efficiency. To ensure adaption to new tasks, possibly different from prior ones, we incorporate our model into gradient boosting, transitioning from a meta-learning driven model towards a specialized one on the current task. Empirical results demonstrate superior performance on both HPO and NAS benchmarks, as well as synthetic benchmarks with heteroscedastic noise.

Despite the promising experimental results, some limitations of the method should be noted. As discussed by Tiao et al. (2021), the threshold parameter $\tau$ in likelihood-free BO algorithms, which controls the exploitation and exploration trade-off, should be treated more carefully. One might consider probabilistic treatment for this hyperparameter. We also observed over-confident predictions in our experiments, for example the Forrester ensemble in Appendix F, where not all runs find the true optimum. This is especially important for problems with more complicated loss landscapes, but mitigation strategies exist.

Directions for future work include extensions to parallel BO with Thompson sampling (Kandasamy et al., 2018), multi-fidelity optimization for HPO problems (Falkner et al., 2018), multi-objective optimization (Hernandez-Lobato et al., 2016), BO with automatic stopping (Makarova et al., 2022) and Bayesian deep active learning (Gal et al., 2017).

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

# A LIKELIHOOD-FREE ACQUISITION FUNCTION

For completeness, we provide the proofs and derivations from Bergstra et al. (2011), Tiao et al. (2021), and Song et al. (2022). Recall from Eq. (1) that the expected utility function is defined as the expectation of the improvement utility function $U(y; \tau)$ over the posterior predictive distribution $p(y \,|\, \boldsymbol{x}, \mathcal{D}_n)$. For the specific expected improvement (EI) acquisition function, where the utility function is $U(y; \tau) := \max(\tau - y, 0)$, the function reads:

$$
\begin{aligned}
\alpha(\boldsymbol{x}; \mathcal{D}_n, \tau) := \mathbb{E}_{p(y \,|\, \boldsymbol{x}, \mathcal{D}_n)} &= \int_{-\infty}^{\infty} U(y; \tau) p(y \,|\, \boldsymbol{x}, \mathcal{D}_n) dy \\
&= \int_{\tau}^{\infty} (\tau - y) p(y \,|\, \boldsymbol{x}, \mathcal{D}_n) dy \\
&= \frac{1}{p(\boldsymbol{x} \,|\, \mathcal{D}_n)} \int_{\tau}^{\infty} (\tau - y) p(\boldsymbol{x} \,|\, y, \mathcal{D}_n) p(y \,|\, \mathcal{D}_N) dy,
\end{aligned}
\tag{8}
$$

We follow the prove from Bergstra et al. (2011) and Tiao et al. (2021) and consider $\ell(\boldsymbol{x}) = p(\boldsymbol{x} \,|\, y \le \tau, \mathcal{D}_n)$ and $g(\boldsymbol{x}) = p(\boldsymbol{x} \,|\, y > \tau, \mathcal{D}_n)$. The denominator of the above equation can then be written as:

$$
\begin{aligned}
p(\boldsymbol{x} \,|\, \mathcal{D}_n) &= \int_{-\infty}^{\infty} p(\boldsymbol{x} \,|\, y, \mathcal{D}_n) p(y \,|\, \mathcal{D}_n) dy \\
&= \ell(\boldsymbol{x}) \int_{-\infty}^{\tau} p(y \,|\, \mathcal{D}_n) dy + g(\boldsymbol{x}) \int_{\tau}^{\infty} p(y \,|\, \mathcal{D}_n) dy \\
&= \gamma \ell(\boldsymbol{x}) + (1 - \gamma) g(\boldsymbol{x}),
\end{aligned}
\tag{9}
$$

where $\gamma := \Phi(\tau) = p(y \le \tau \,|\, \mathcal{D}_n)$. The numerator can be evaluated as:

$$
\int_{\tau}^{\infty} \max(\tau - y, 0) p(\boldsymbol{x} \,|\, y, \mathcal{D}_n) p(y \,|\, \mathcal{D}_N) dy = \ell(\boldsymbol{x}) \int_{\tau}^{\infty} \max(\tau - y, 0) p(y \,|\, \mathcal{D}_n) dy
\tag{10}
$$

$$
= \ell(\boldsymbol{x}) \tau \int_{\tau}^{\infty} p(y \,|\, \mathcal{D}_n) dy - \ell(\boldsymbol{x}) \int_{\tau}^{\infty} y p(y \,|\, \mathcal{D}_n) dy
\tag{11}
$$

$$
= \gamma \tau \ell(\boldsymbol{x}) - \ell(\boldsymbol{x}) \int_{\tau}^{\infty} y p(y \,|\, \mathcal{D}_n) dy
\tag{12}
$$

$$
= K \cdot \ell(\boldsymbol{x}),
\tag{13}
$$

where $K = \gamma \tau - \int_{\tau}^{\infty} y p(y \,|\, \mathcal{D}_n) dy$. Therefore the EI acquisition function is equivalent to the $\gamma$-relative density ratio up to a constant $K$,

$$
\underbrace{\alpha(\boldsymbol{x}; \mathcal{D}_N, \tau)}_{\text{expected improvement}} \propto \underbrace{\frac{\ell(\boldsymbol{x})}{\gamma \ell(\boldsymbol{x}) + (1 - \gamma) g(\boldsymbol{x})}}_{\text{density ratio}}
\tag{14}
$$

Intuitively, one can think of the configuration $\boldsymbol{x}$ whose corresponding $y \le \tau$ as *good* configurations, and the those with $y > \tau$ are *bad* configurations. Then the density ratio can be interpreted as the ratio between the model's belief on that configuration being a good and a bad configuration.

For Tree-structured Parzen Estimators (TPE, Bergstra et al. (2011)), they first select the $\gamma$ as hyperparameter and estimate this density ratio by explicitly modelling $\ell(\boldsymbol{x})$ and $g(\boldsymbol{x})$ using kernel density estimation. As for BORE by Tiao et al. (2021), they model the density ratio by class probability, where $\ell(\boldsymbol{x}) = p(\boldsymbol{x} \,|\, k = 1)$ and $g = p(\boldsymbol{x} \,|\, k = 0)$.

Song et al. (2022) proof that density ratio acquisition function is not always equivalent to EI. Bergstra et al. (2011) and Tiao et al. (2021) claims that Eq. (10) hold true, which is $\ell(\boldsymbol{x})$ times a value independent of **x**. Bergstra et al. (2011) assumes $\ell(\boldsymbol{x})$ is independent of $y$ once $y \le \tau$ and therefore we can take $\ell(\boldsymbol{x})$ directly out of the integral.

Nevertheless, $p(\mathbf{x} \,|\, y \le \tau, \mathcal{D}_n)$ is still dependent on $y$ even if $y \le \tau$, because it is a conditional probability condition on $y$. They confused with depending on $y$ and depending on $y \le \tau$, where these

two statements are different and mean differently. By just satisfying the condition $y \le \tau$, it does not necessarily mean that it become independent on $y$. From the definition of conditional probability:

$$p(\boldsymbol{x} \mid y \le \tau, \mathcal{D}_n) = \frac{\int_{-\infty}^{\tau} p(\boldsymbol{x}, y \mid \mathcal{D}_n) dy}{\int_{-\infty}^{\tau} p(y \mid \mathcal{D}_n) dy} \ne p(\mathbf{x} \mid y, \mathcal{D}_n), \tag{15}$$

they are not equivalent. Intuitively, to understand this, is for example, the probability of the configuration $\boldsymbol{x}$ still depends on the its corresponding value $y$ even though $y < \tau$. Another interpretation for this would be, density ratio acquisition function would treat all $(\boldsymbol{x}, y)$ pairs above it with equal importance, because it does not depend on $y$ any more if $y \le \tau$ according to the independence assumption. In fact, expected improvement weighted the importance of $(\boldsymbol{x}, y)$ pairs by how much $y$ is lower than $\tau$.

To tackle this issue, Song et al. (2022) propose to estimate the density ratio via variational f-divergence estimation (Nguyen et al., 2010). They provide a variational representation for the expected utility function at any point $\boldsymbol{x}$, provided the samples from some distribution $p(y \mid \boldsymbol{x})$, which can replace the integration with a variational objective function:

$$\mathbb{E}_{p(y|\boldsymbol{x})}[U(y;\tau)] = \arg\max_{s \in [0,\infty)} \mathbb{E}_{p(y|\boldsymbol{x})}[U(y;\tau)f'(s) - f^*(f'(s))], \tag{16}$$

where the utility function $U$ is non-negative, and $f : [0,\infty) \to \mathbb{R}$ is a strictly convex function, and $f^*$ is the convex conjugate of $f$. This acquisition function does not model distributions with probability but only samples from the observations $D_n$.

They consider their acquisition function $\alpha^{\text{LFBO}} = \hat{S}_{\mathcal{D}_{n,\tau}}(\boldsymbol{x})$, and state that the acquisition function can be written as:

$$\hat{S}_{\mathcal{D}_{n,\tau}}(\boldsymbol{x}) = \arg\max \mathbb{E}_{\mathcal{D}_n}[U(y;\tau)f'(S(\boldsymbol{x})) - f^*(f'(S(\boldsymbol{x})))]. \tag{17}$$

This means by optimizing a variaitional objective in the search space $\mathcal{X}$, we can recover an expected utility acquisition function over $\boldsymbol{x}$, which makes $\mathcal{L}^{\text{LFBO}}$ equivalent to expected utility acquisition function. For practical purpose, they choose a specific convex funciton $f$: $f(r) = r \log \frac{r}{r+1} + \log \frac{1}{r+1}$ for all $r > 0$. For their acquisition function $\alpha^{\text{LFBO}}$ they consider:

$$\alpha^{\text{LFBO}}(\boldsymbol{x}; \mathcal{D}_n, \tau) = \hat{C}_{\mathcal{D}_{n,\tau}}(\boldsymbol{x})/(1 - \hat{C}_{\mathcal{D}_{n,\tau}}(\boldsymbol{x})), \tag{18}$$

where $\hat{C}_{\mathcal{D}_{n,\tau}}$ is the maximizer of some objecetive over $C : \mathcal{X} \to (0,1)$. By applying this $f$ and Eq. (18) to Eq. (17), the objective for maximization objective $C$ become:

$$\mathbb{E}_{(\boldsymbol{x},y) \sim p(\boldsymbol{x}, y \mid \mathcal{D}_N)}[U(y;\tau) \log C(\boldsymbol{x}) + \log(1 - C(\boldsymbol{x}))]. \tag{19}$$

## B   PROBIT APPROXIMATION

Let's set $a = \boldsymbol{z}^T \boldsymbol{\phi} + m(\boldsymbol{\phi})$, and the distribution of $a$ would be a Gaussian $\mathcal{N}(a \mid \mu_a, \sigma_a)$, where $\mu_a$ and $\sigma_a$ are respectively:

$$\mu_a = \mathbb{E}[a] = \int p(a) a \, da = \int q(\boldsymbol{z})(\boldsymbol{z}^T \boldsymbol{\phi} + m(\boldsymbol{\phi})) \, d\boldsymbol{z} = \boldsymbol{z}_{\text{MAP}}^T \boldsymbol{\phi} + m(\boldsymbol{\phi}) \tag{20}$$

$$\begin{aligned}
\sigma_a^2 = \text{var}[a] &= \int p(a)\{a^2 - \mathbb{E}[a]^2\} \, da \\
&= \int q(\boldsymbol{z})\{(\boldsymbol{z}^T \boldsymbol{\phi} + m(\boldsymbol{\phi}))^2 - (\mathbf{m}_N^T \boldsymbol{\phi} + m(\boldsymbol{\phi}))^2\} \, d\boldsymbol{z} \\
&= \boldsymbol{\phi}^T \boldsymbol{\Sigma}_N \boldsymbol{\phi}
\end{aligned} \tag{21}$$

Thus our approximation to the predictive distribution becomes:

$$p(k = 1 \mid g(\boldsymbol{x}), \mathcal{D}_N) = \int \sigma(a) \mathcal{N}(a \mid \mu_a, \sigma_a^2) \, da \tag{22}$$

To evaluate the integral in Eq. (22), we can obtain a good approximation by making use of the close similarity between the logistic sigmoid function $\sigma(a)$ and the probit function, which is given by the cumulative distribution of the standard Gaussian $\Phi(a) = \int_{-\infty}^a \mathcal{N}(\theta \mid 0, 1) \, d\theta$:

$$\int \Phi(\lambda a) \mathcal{N}(a \mid \mu, \sigma^2) \, da = \Phi\left(\frac{\mu}{(\lambda^{-2} + \sigma^2)^{1/2}}\right) \tag{23}$$

We apply the approximation $\sigma(a) \simeq \Phi(\lambda a)$ to the probit functions appearing on both sides of the equation:

$$\int \sigma(a) \mathcal{N}(a \mid \mu, \sigma^2) \, da \simeq \sigma((1 + \pi \sigma^2 / 8)^{-1/2} \mu) \tag{24}$$

Therefore we obtain predictive distribution in the form:

$$p(k = 1 \mid g(\boldsymbol{x}), \mathcal{D}_N) \simeq \sigma((1 + \pi \sigma_a^2 / 8)^{-1/2} \mu_a) \tag{25}$$

where $\mu_a$ is Eq. (20) and $\sigma_a^2$ is Eq. (21).

# C RUNTIME ANALYSIS

The experiments in Section 5 show the performance over optimization steps. To be complementary, we demonstrate the same results from a different perspective, namely, we report the immediate regrets as a function of estimated wall-clock time. To obtain the realistic wall-clock time, we accumulate the time to optimize for corresponding BO methods and the recorded runtime for configurations in the benchmarks. Notice that, all the methods run for the same number of steps in an experiment.

As the results are shown in Figs. 7–9, MALIBO and its variants attain the best warm-starting performance across all benchmarks and constantly achieve the lowest regrets with the same amount of time. LFBO and BORE are two competitive methods in terms of end performance, but both need quite some time to catch up the regrets of MALIBO and its variants. GC3P is the method with closest time performance as MALIBO in most of the benchmarks. However, their performance is not as stable as MALIBO, especially for NASBench201 and some of the MLBench problems, where the meta-learning fail to warm-start the optimization. Similarly, the meta-learning of RGPE and ABLR do not deliver any advantage over the non-meta-learning baselines and thus end up with close performance as normal GP.

To further investigate the time efficiency of MALIBO, we illustrates the runtime of the optimization algorithms for each step in Fig. 10. The runtime for MALIBO and its variants is the fastest among all the meta-learning methods, while only slightly slower than the non-meta-learning likelihood-free methods, namely BORE and LFBO. Due to the increasing amount of observations, the runtime of almost all the methods grow over iterations, especially for RGPE and GP, where the growths are the most significant. Although ABLR and GC3P are around a order of magnitude slower than MALIBO at the beginning, but their runtimes remain stable over steps.

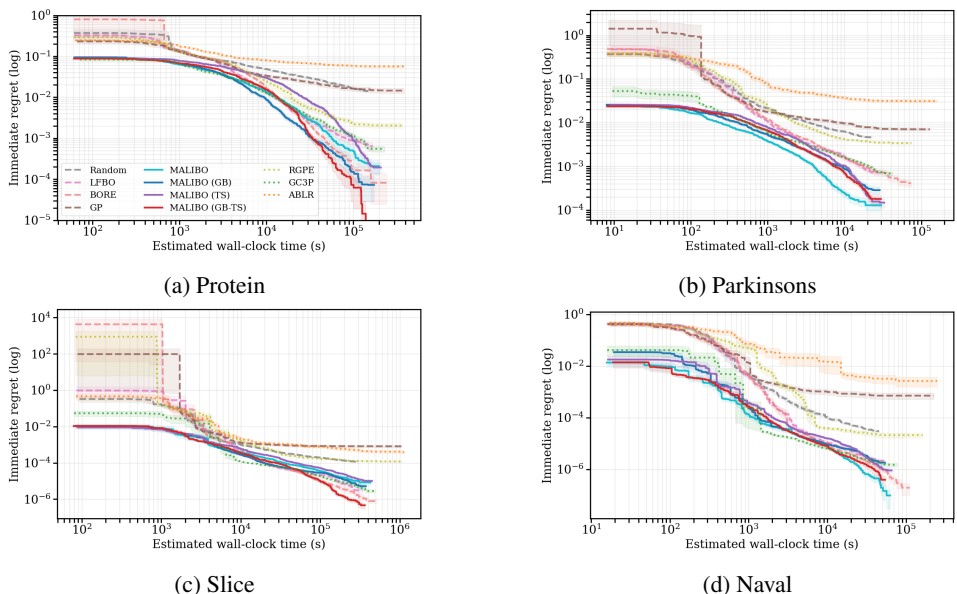

Figure 7: Immediate regrets of different BO algorithms on the HPOBench neural network tuning problem. Each algorithm runs for 500 iterations and we show the corresponding estimated wall-clock time on the $x$ axis in log scale.

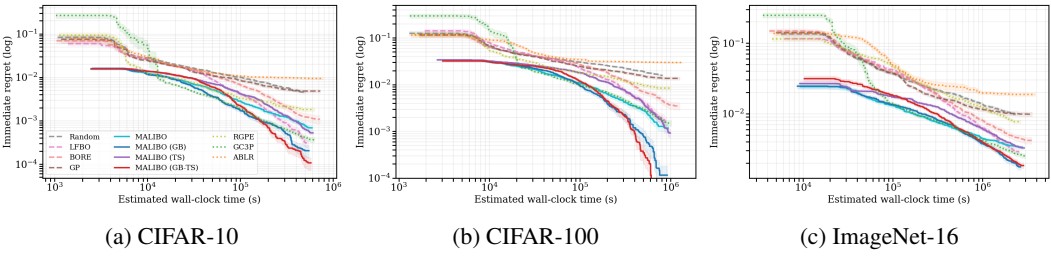

(a) CIFAR-10      (b) CIFAR-100      (c) ImageNet-16

Figure 8: Immediate regrets of different BO algorithms on the NASBench201 neural network architecture search problem. Each algorithm runs for 200 iterations and we show the corresponding estimated wall-clock time on the $x$ axis in log scale.

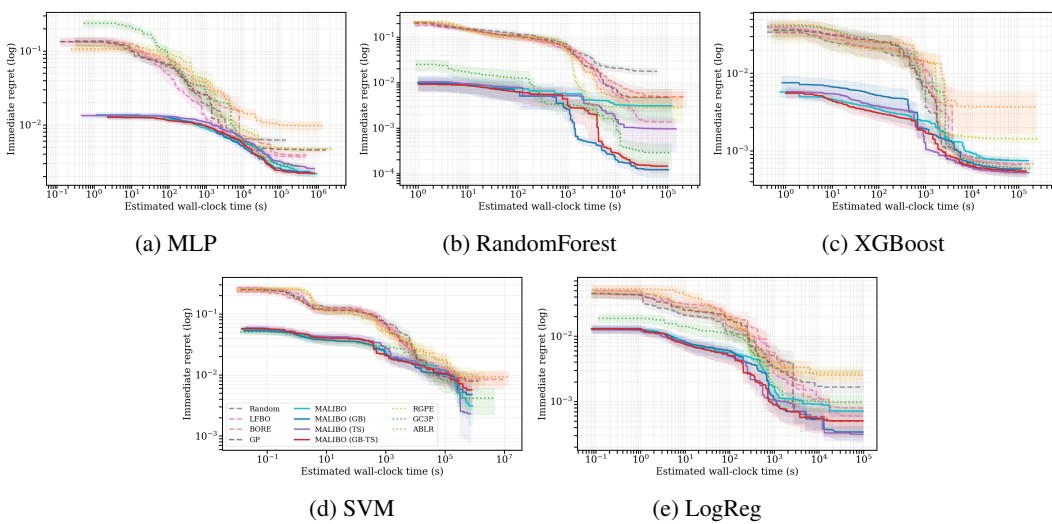

(a) MLP      (b) RandomForest      (c) XGBoost

(d) SVM      (e) LogReg

Figure 9: Immediate regrets of different BO algorithms on the HPOBench hyperparameter tuning for machine learning algorithms. Each algorithm runs for 72 iterations and we show the corresponding estimated wall-clock time on the $x$ axis in log scale.

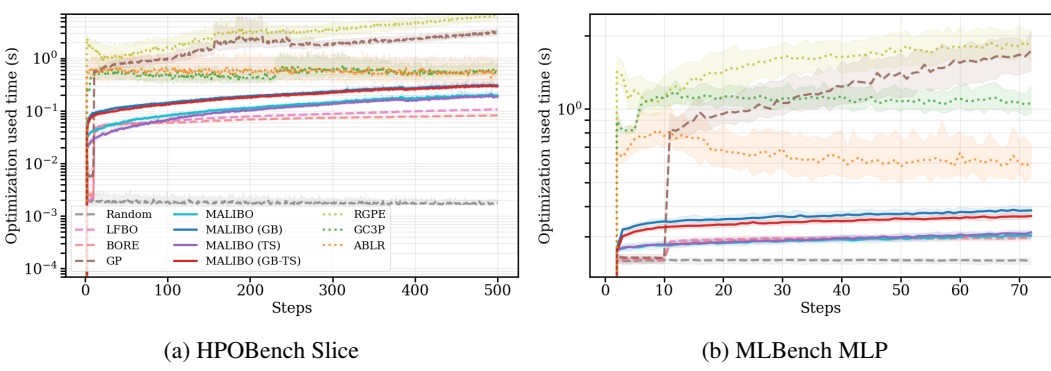

(a) HPOBench Slice      (b) MLBench MLP

Figure 10: Runtime of different BO algorithms over optimization steps. We show the typical results for two benchmarks and plot the medial inter-quantiles to remove outliers.

# D ABLATION STUDIES

We conduct ablation studies to show how the different components of MALIBO help for the optimization. To be specific, we first demonstrate the meta-learned features representation for two different function in Appendix D.1 to provide an intuitive visualization for the expressiveness of our meta-learning model. To answer the questions of what components and how they help the optimization, we show the experiments of MALIBO and its variants running on 4 different synthetic function benchmarks without any meta-learning in Appendix D.2.

## D.1 LATENT FEATURE ANALYSIS

We show how our meta-learning model learns a feature representation from meta-data. The latent features $\phi$ can be considered as basis functions and are supposed to represent the structure of the meta-data distribution. With the properly learned features $\phi$, the mean layer and the task embedding layer $z$ can combine them to obtain a function with similar structure to the meta-data while matching the shape of target function.

In order to learn a effective feature representation, one should capture both the local and global structure of the function. Therefore, we select two types of function to study the effectiveness of features learning for MALIBO: i) Forrester functions (Sobester et al., 2008) with two very likely positions for the global optimum, which has rich local structure. ii) quadratic functions, where the functions share a certain global shape, but the optima could be located anywhere in the search space. For more details on the synthetic functions and the generation of meta-data, we refer to Appendix F.

The results for these two synthetic functions are shown in Fig. 11 and Fig. 12 respectively. On one hand, we can see that in Fig. 11 the features learned by MALIBO has either maximum or minimum around the two likely optima, which means the model successfully infers the local structure from the meta-data. On the other hand, Fig. 12 shows that, even without a clear location of optima, the features still learns the shape of quadratic functions.

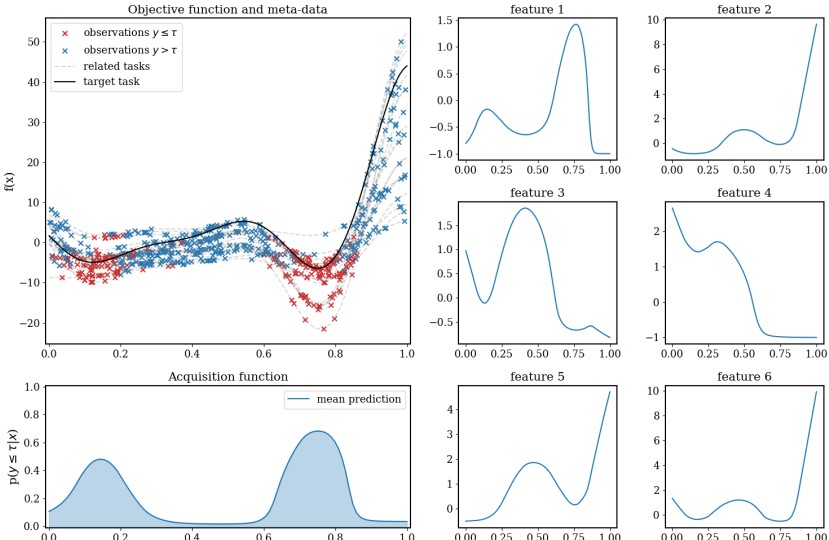

Figure 11: Left: Forrester functions with two very likely optima as target function and related-tasks. The learned acquisition function is shown below. Right: Meta-learned latent features from related-tasks. The latent features show the model successfully infer the location of two optima, resulting in a acquisition with two modes around the same locations.

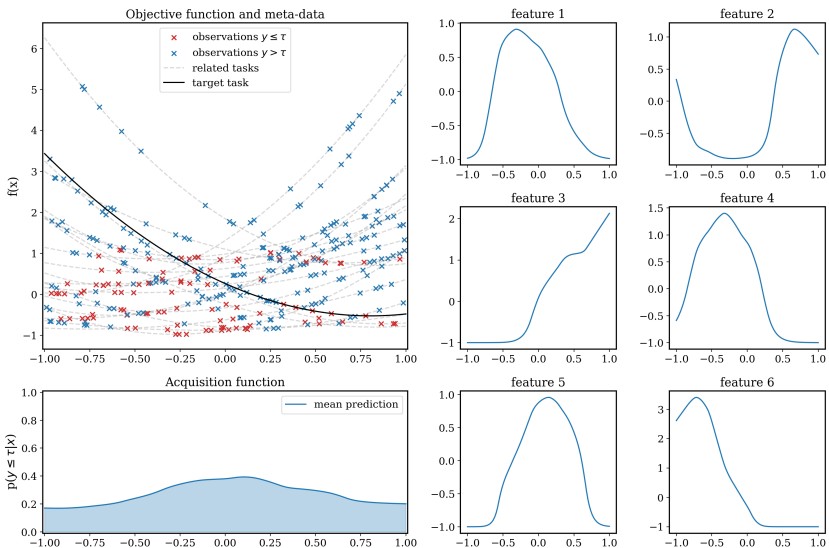

Figure 12: Left: Quadratic functions with varying optima as target function and related-tasks. The learned acquisition function is shown below. Right: Meta-learned latent features from related-tasks. The latent features show the model can learn the global structure shared across all the meta-data, even though there is no clear position for optima.

## D.2 EFFECTS OF GRADIENT BOOSTING AND THOMPSON SAMPLING

The vanilla MALIBO uses Bayesian logistic regression for task adaptation, which leverages the meta-learned feature. However, the performance depends heavily on the quality of the data and the latent features learned from the data. In practice, the amount and the quality of the data are often not guaranteed. We introduce gradient boosting as a residual model to safeguard the optimization when little meta-data or a large discrepancy between the training data and the meta-data distribution exists. Further, we apply Thompson sampling to encourage exploration, which enables MALIBO to collect more information about the target function by exploring the search space efficiently. To show the usefulness of these components in MALIBO, we remove the effects of meta-learning by optimizing the target synthetic functions without any meta-training, and therefore the experiments will focus only on the effects on the gradient boosting and Thompson sampling.

We use four synthetic benchmarks for the experiments, namely quadratic, Forrester, Branin and Hartmann3 functions. We refer to Appendix F for more details of the synthetic functions. As the results shown in Fig. 13, the MALIBO (GB) variant consistently demonstrate better performance than the vanilla MALIBO except in Forrester, where the non-meta-learning MALIBO can easily stuck in one local optima. This indicates that, when there is little meta-data, gradient boosting can help the model to converge toward a lower regret and the Bayesian logistic regression fail to optimize. For MALIBO (TS), the results show the model achieve lower immediate regrets than vanilla MALIBO across all the benchmark, because it encourages the exploration in the search space. However, due to the fact that Thompson sampling improve only exploration, for example Branin and Hartmann3 function, the MALIBO (GB) outperforms the MALIBO (GB) variant. Last but not least, the results in experiments show that, by combing the exploitation of gradient boosting and exploration of Thompson sampling, MALIBO (GB-TS) achieves the best performance across all benchmarks.

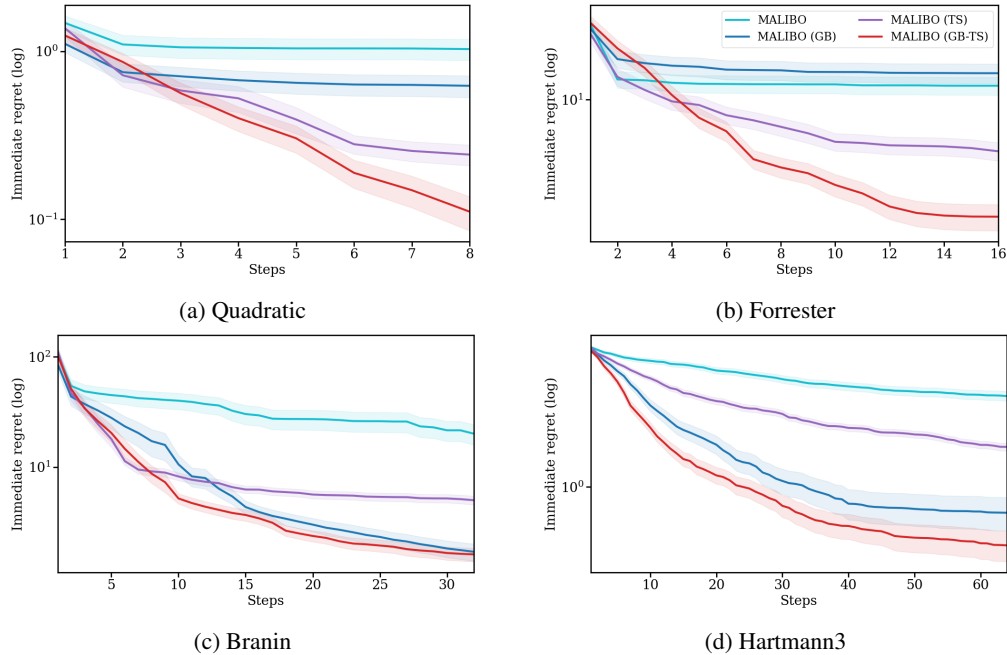

(a) Quadratic

(b) Forrester

(c) Branin

(d) Hartmann3

Figure 13: Results of MALIBO and its variants on four synthetic function benchmarks without meta-training. We report the standard error of immediate regrets over 100 runs.

# E  STEP-THROUGH VISUALIZATION

For illustration purposes, we provide step-through visualizations on a Forrester and a quadratic function. For details of the synthetic functions, we refer to Appendix H.5 and Appendix H.4 respectively. We use the same meta-trained model for the visualizations as the one used in Appendix D.1 for the corresponding problem.

We demonstrate the advantage of using Thompson sampling in two parts. First, by showing the MALIBO (TS) variant optimize functions sequentially, which is in correspondence to normal BO pipeline, we can see how the algorithm explore the space efficiently. The illustrations are demonstrated in Fig. 14 and Fig. 15. Thereafter, we show two toy examples of synchronous parallel BO (Kandasamy et al., 2018) using MALIBO (TS) on the same functions. To be specific, we use three Thompson samples as acquisition functions in each iteration, and evaluates the three proposed points for the next optimization step. We demonstrate that, MALIBO (TS) can be easily extended to parallel BO with the help of Thompson sampling.

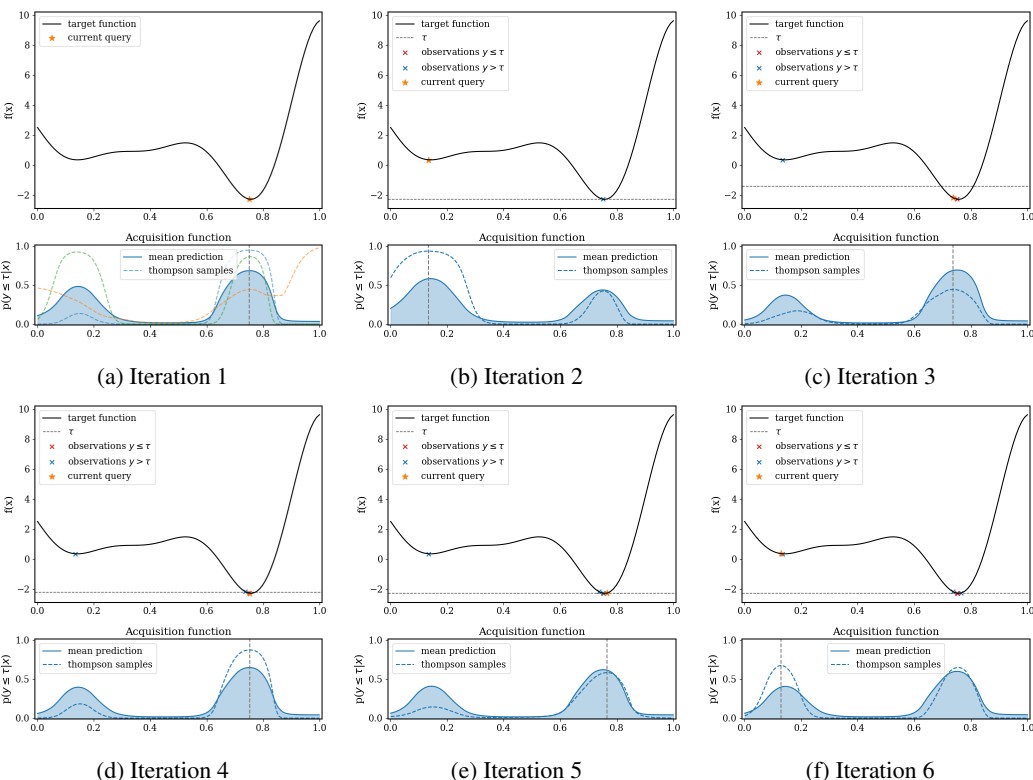

Figure 14: Sequential Thompson sampling with MALIBO optimizing a Forrester function. We show the mean prediction together with the Thompson samples of the acquisition function in the lower part of each sub-figure. At the first iteration, MALIBO (TS) picks the point with highest mean prediction of the acquisition function, which is already close to the global optimum. Thereafter, it chooses the point according to the maximum prediction of the Thompson sample, where it explored another location of interest on the left-hand side. After the second step, the model believes that the promising region is on the right-hand side and start exploiting that region, where the true optimum locates. Notice that, almost all the Thompson samples remain the shape close to the mean prediction of acquisition function, which means it will explore the search space according to the current belief rather than simply random samples.

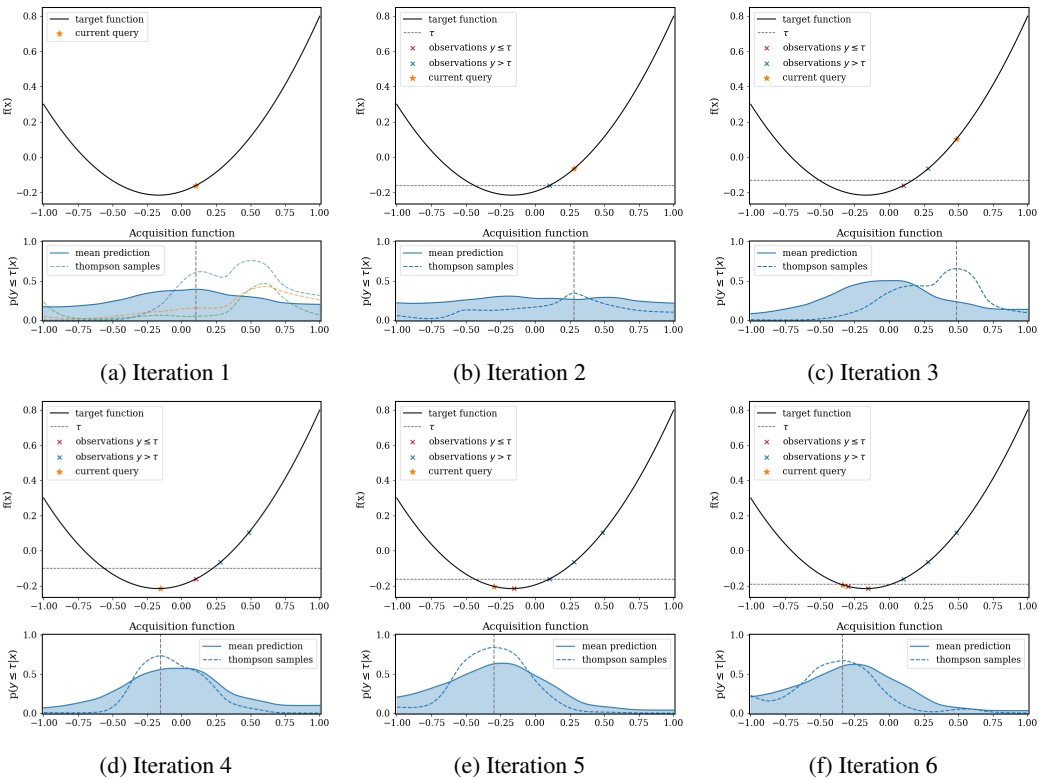

Figure 15: Sequential Thompson sampling with MALIBO optimizing a quadratic function. We show the mean prediction together with the Thompson samples of the acquisition function in the lower part of each sub-figure. Since the meta-data of quadratic functions do not have a distinctive promising region but only sharing a similar global structure, the first point MALIBO (TS) picks is not informative for the problem. However, after exploring the search space with Thompson samples of the acquisition function, MALIBO (TS) gradually forms a belief that the optimum locates around $x = -0.25$. Notice that, even though the meta-learned acquisition function is not informative about the location of the optimum, but the shape of the mean prediction and the Thompson samples of the acquisition function remain similar to a quadratic function over iterations, which conforms the global features learned from the meta-data as shown in Fig. 12.

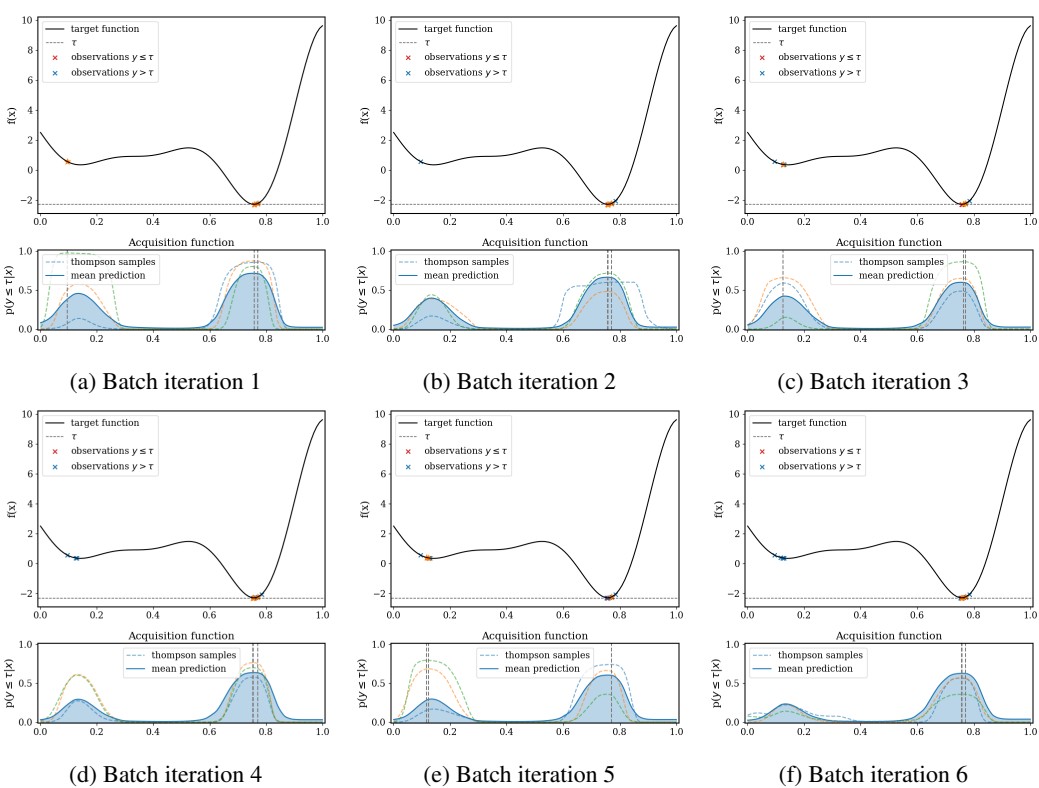

Figure 16: Synchronous parallel Thompson sampling using MALIBO (TS) to optimize a Forrester function. Every iteration we draw three samples as acquisition functions and utilize the resulting query points as observations for the next optimization step. In the first iteration, MALIBO (TS) already acquires three observations which covers both the likely positions for the optimum thanks to the parallelism. Subsequently, MALIBO (TS) exploits more often around the area where the true optimum locates than exploring the other area of interest. At the last iteration, all of the three Thompson samples have already been skew toward the right-hand side, which shows MALIBO (TS) converges to the correct region.

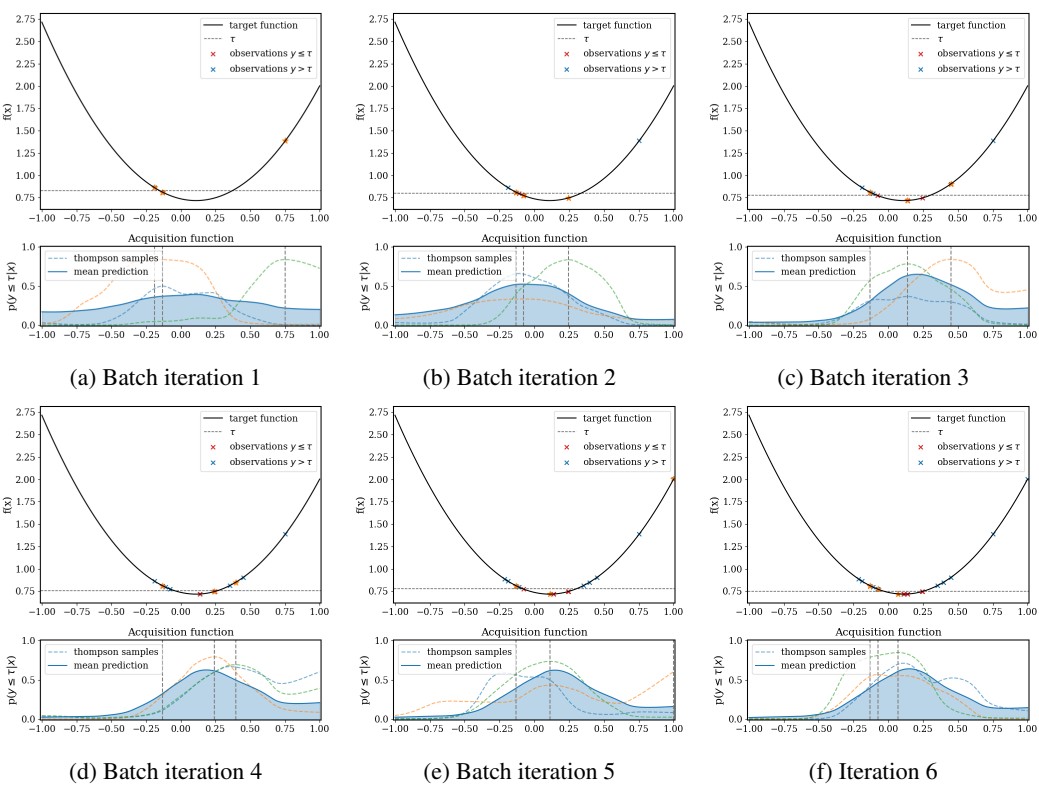

Figure 17: Synchronous parallel Thompson sampling using MALIBO (TS) to optimize a quadratic function. Every iteration we draw three samples as acquisition functions and utilize the resulting query points as observations for the next optimization step. The exploration from the Thompson sampling helps to discover the boundary of the promising region, i.e. the observations $y \leq \tau$ (blue crosses), in fewer iterations and the range of promising region keep reducing as more unfavorable points close to the true optimum were acquired.

## F    ADDITIONAL BENCHMARKS

In this section we show more results for experiments with multiplicative noise as in Section 5. With the settings remaining the same, we perform our experiments on Forester (Sobester et al., 2008) and Hartmann3 (Dixon, 1978) function ensembles. We refer to Appendices H.5 and H.7 respectively for more details.

**Forrester function ensemble**    For meta-training in Forester ensemble experiment, we randomly sampled noisy 32 observations in 64 prior-tasks. The results is shown in Fig. 18. MALIBO and its variants keep showing strong warm-starting performance and stay robust to noise compared to other methods. However, all the likelihood-free BO methods, namely LFBO, MALIBO and its variants, seem to stuck in local minimum in some runs, resulting in almost no improvement over the optimization process. The performance of all MALIBO variants is on-par with GC3P in all cases. Although most of the GP-based methods, namely GP, RGPE and ABLR, all outperform the other likelihood-free based methods, however after increasing the noise level, their performances degrade significantly.

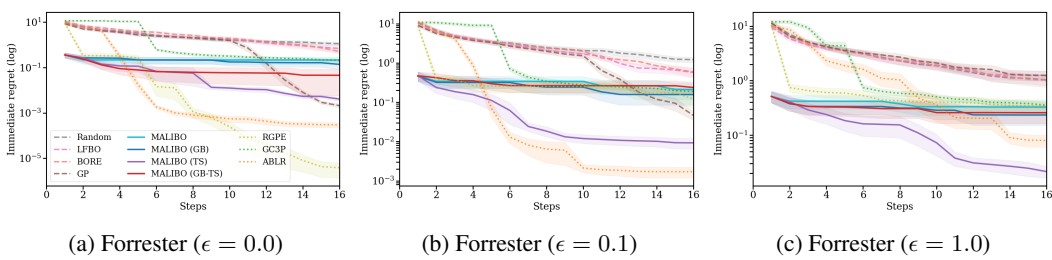

(a) Forrester ($\epsilon = 0.0$)          (b) Forrester ($\epsilon = 0.1$)          (c) Forrester ($\epsilon = 1.0$)

Figure 18: Immediate regret for different BO algorithms on Forrester function ensembles ($D = 1$) with different levels of multiplicative noise.

**Hartmann3 function ensemble**    We randomly sampled noisy 512 observations in 256 prior-tasks in the Hartmann3 ensemble experiment. The results is shown in Fig. 19. All MALIBO variants shows the strongest meta-learning performance in all settings, and the gradient boosting variants report the lowest regrets together with ABLR in noise-free function. All MALIBO variants remain robust to noise, especially the vanilla MALIBO. The GP-based method, although they have strong performance in the noise-free case, especially RGPE, but they degrade significantly after the noise level increased.

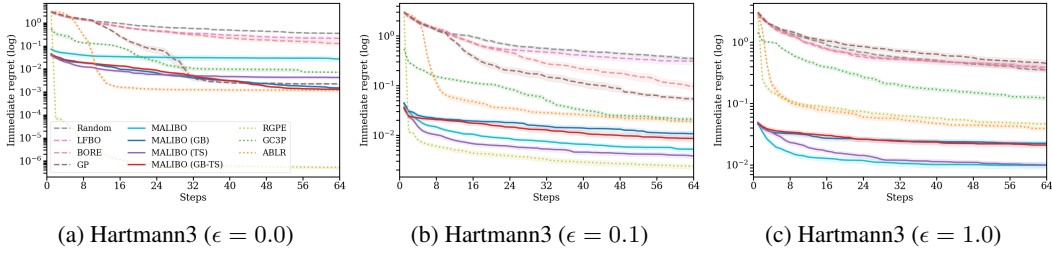

(a) Hartmann3 ($\epsilon = 0.0$)          (b) Hartmann3 ($\epsilon = 0.1$)          (c) Hartmann3 ($\epsilon = 1.0$)

Figure 19: Immediate regret for different BO algorithms on Hartmann3 function ensembles ($D = 3$) with different levels of multiplicative noise.

## G  EXPERIMENTAL DETAILS

Consider a Residual Feed Forward Network (ResFFN) (He et al., 2016) architecture ResFFN-4-64, which contains 4 residual feedforward layers with 64 units. We use ResFFN-4-64 to learn the latent feature representation, with 4 hidden layers, each with 64 units. For the mean layer $m(\cdot)$ and task embedding layer $z$, we use a fully connected layer with 50 units for each. We use ELU (Clevert et al., 2016) as activation function for our problem following Tiao et al. (2021).

During meta-training, we optimize the weights with ADAM (Kingma & Ba, 2015) using batch size of $B = 256$, and polynomial decay for learning rate, with the initial learning rate $lr_{initial} = 10^{-3}$, end learning rate $lr_{final} = 2^{-4}$ and the exponent set to 2. The model is trained for 2048 epochs with early stopping. We set the regularization factor $\lambda = 0.1$ in Eq. (4) and follow the approach in (Berkenkamp et al., 2021) to estimate the weights for modified Kolmogorov-Smirnov test and covariance regularization. In task adaptation, we optimize the task embedding with L-BFGS (Byrd et al., 1995).

For the gradient boosting applied to BORE, LFBO and MALIBO, we use the implementation in scikit-learn (Pedregosa et al., 2011) with default settings. The only difference is that we use the meta-learned MALIBO classifier as the initial estimator for gradient boosting variants of MALIBO.

# H  DETAILS OF BENCHMARKS

## H.1  HPOBENCH

The hyperparameters for HPOBench and their ranges are demonstrated in Table 1. All hyparameters are discrete and there are in total 66,208 possible combinations. More details can be found in Klein & Hutter (2019).

Table 1: Configuration spaces for HPOBench

| Hyperparameter | Range |
|---|---|
| Initial LR | $\{ 5 \times 10^{-4}, 1 \times 10^{-3}, 5 \times 10^{-3}, 1 \times 10^{-2}, 5 \times 10^{-2}, 1 \times 10^{-1} \}$ |
| LR Schedule | $\{$ cosine, fixed $\}$ |
| Batch size | $\{ 2^3, 2^4, 2^5, 2^6 \}$ |
| Layer 1 Width | $\{ 2^4, 2^5, 2^6, 2^7, 2^8, 2^9 \}$ |
| Activation | $\{$ relu, tanh $\}$ |
| Dropout rate | $\{ 0.0, 0.3, 0.6 \}$ |
| Layer 2 Width | $\{ 2^4, 2^5, 2^6, 2^7, 2^8, 2^9 \}$ |
| Activation | $\{$ relu, tanh $\}$ |
| Dropout rate | $\{ 0.0, 0.3, 0.6 \}$ |

## H.2  ML ALGORITHMS IN HPOBENCH

The hyperparameters for machine learning (ML) algortihms in HPOBench (Eggensperger et al., 2021) and their ranges are summarized in Table 2. More details can be found in Eggensperger et al. (2021).

Table 2: Configuration spaces for ML algorithms in HPOBench

| Benchmark | Hyperparameter | type | Log | Range |
|---|---|---|---|---|
| SVM | C | float | ✓ | $[ 2^{-10}, 2^{10} ]$ |
| | gamma | float | ✓ | $[ 2^{-10}, 2^{10} ]$ |
| LogReg | alpha | float | ✓ | $[ 1e^{-5}, 1.0 ]$ |
| | eta0 | float | ✓ | $[ 1e^{-5}, 1.0 ]$ |
| XGBoost | colsample_bytree | float | ✗ | $[ 0.1, 1.0 ]$ |
| | eta | float | ✓ | $[ 2^{-10}, 1.0 ]$ |
| | max_depth | int | ✓ | $[ 1, 50 ]$ |
| | reg_lambda | float | ✓ | $[ 2^{-10}, 2^{-10} ]$ |
| RandomForest | max_depth | int | ✓ | $[ 1, 50 ]$ |
| | max_features | float | ✗ | $[ 0.0, 1.0 ]$ |
| | min_samples_leaf | int | ✗ | $[ 1, 2 ]$ |
| | min_samples_split | int | ✓ | $[ 2, 128 ]$ |
| MLP | alpha | float | ✓ | $[ 1.0e^{-8}, 1.0 ]$ |
| | batch_size | int | ✓ | $[ 4, 256 ]$ |
| | depth | int | ✗ | $[ 1, 3 ]$ |
| | learning_rate_init | float | ✓ | $[ 1.0e^{-5}, 1.0 ]$ |
| | width | int | ✓ | $[ 16, 1024 ]$ |

## H.3  NASBENCH201

The hyperparameters for NASBench201 and their ranges are summarized in Table 3. All hyparameters are discrete and there are in total 15,625 possible combinations. More details can be found in Dong & Yang (2020).

Table 3: Configuration spaces for NASBench201

| Hyperparameter | Range |
| --- | --- |
| ARC 0 | { none, skip-connect, conv-1 $\times$ 1, conv-3 $\times$ 3, avg-pool-3 $\times$ 3 } |
| ARC 1 | { none, skip-connect, conv-1 $\times$ 1, conv-3 $\times$ 3, avg-pool-3 $\times$ 3 } |
| ARC 2 | { none, skip-connect, conv-1 $\times$ 1, conv-3 $\times$ 3, avg-pool-3 $\times$ 3 } |
| ARC 3 | { none, skip-connect, conv-1 $\times$ 1, conv-3 $\times$ 3, avg-pool-3 $\times$ 3 } |
| ARC 4 | { none, skip-connect, conv-1 $\times$ 1, conv-3 $\times$ 3, avg-pool-3 $\times$ 3 } |
| ARC 5 | { none, skip-connect, conv-1 $\times$ 1, conv-3 $\times$ 3, avg-pool-3 $\times$ 3 } |

### H.4 THE QUADRATIC ENSEMBLE

The function for the quadratic ensemble is defined as:

$$f(x, a, b, c) = (a \cdot (x - b))^2 - c \qquad x \in [0, 1] \tag{26}$$

To form the ensemble, we choose the distribution for the parameters as:

$$a \sim \mathcal{U}(0.5, 1.5) \quad b \sim \mathcal{U}(-0.9, 0.9) \quad c \sim \mathcal{U}(-1, 1) \tag{27}$$

This distribution of parameters ensures that the search space contains the minimum of the quadratic function at $x^* = b$ with $f(x^*) = c$. The location of the optimum has a broad distribution over the function space, which is intended to highlight algorithms that learn the global structure of the ensemble rather than restricting on some small regions of interest.

### H.5 THE FORRESTER ENSEMBLE

The original Forrester function (Sobester et al., 2008) is defined following:

$$f(x, a, b, c) = a \cdot (6x - 2)^2 \sin(12x - 4) + b(x - 0.5) - c \qquad x \in [0, 1] \tag{28}$$

The function has one local and one global minimum, and a zero-gradient inflection point in the domain $x \in [0, 1]$. To form the ensemble, we choose the distribution for the parameters as:

$$a \sim \mathcal{U}(0.2, 3) \quad b \sim \mathcal{U}(-5, 15) \quad c \sim \mathcal{U}(-5, 5) \tag{29}$$

Let $\tau = \{a, b, c\}$ and $p(\tau)$ is a three dimensional uniform distribution. The ranges are chosen around the usually used fixed values for the parameters, namely $a = 0.5$, $b = 10$, $c = -5$.

### H.6 THE BRANIN ENSEMBLE

The function for the Branin ensemble is the following:

$$f(x, a, b, c) = a(x_2 - bx_1^2 + cx_1 - r) + s(1 - t)\cos(x_1) + s \qquad x_1 \in [-5, 10], x_2 \in [0, 15] \tag{30}$$

The distribution for the parameters are chosen as:

$$\begin{aligned} a &\sim \mathcal{U}(0.5, 1.5) \quad b \sim \mathcal{U}(0.1, 0.15) \quad c \sim \mathcal{U}(1.0, 2.0) \\ r &\sim \mathcal{U}(5.0, 7, 0) \quad s \sim \mathcal{U}(8.0, 12.0) \quad t \sim \mathcal{U}(0.03, 0.05) \end{aligned} \tag{31}$$

Let $\tau = \{a, b, c, r, s, t\}$ and $p(\tau)$ is a six dimensional uniform distribution. The ranges are chosen around the usually used fixed values for the parameters, namely $a = 1$, $b = 5.1/(4\pi^2)$, $c = 5/\pi$, $r = 6$, $s = 10$ and $t = 1/(8\pi)$.

### H.7 THE HARTMANN3 ENSEMBLE

The function for Hartmann3 (Dixon, 1978) ensemble reads:

$$f(x, \alpha_1, \alpha_2, \alpha_3, \alpha_4) = -\sum_{i=1}^{4} \alpha_i \exp\left(-\sum_{j=1}^{3} A_{i,j}(x_j - P_{i,j})^2\right) \qquad x \in [0, 1]$$

$$\boldsymbol{A} = \begin{bmatrix} 3.0 & 10 & 30 \\ 0.1 & 10 & 35 \\ 3.0 & 10 & 30 \\ 0.1 & 10 & 35 \end{bmatrix} \qquad \boldsymbol{P} = 10^{-4} \cdot \begin{bmatrix} 3689 & 1170 & 2673 \\ 4699 & 4387 & 7470 \\ 1091 & 8732 & 5547 \\ 381 & 5743 & 8828 \end{bmatrix} \tag{32}$$

To form the ensemble, we choose the distribution for the parameters as:

$$\alpha_1 \sim \mathcal{U}(0.0, 2.0) \quad \alpha_2 \sim \mathcal{U}(0.0, 2.0) \quad \alpha_3 \sim \mathcal{U}(2.0, 4.0) \quad \alpha_4 \sim \mathcal{U}(2.0, 4.0) \tag{33}$$

Let $\tau = \{\alpha_1, \alpha_2, \alpha_3, \alpha_4\}$ and $p(\tau)$ is a four dimensional uniform distribution.

