# OpenReview forum: "MALIBO: Meta-Learning for Likelihood-free Bayesian Optimization"
_ICLR.cc/2023/Conference — Submitted to ICLR 2023_

### Official Review · Reviewer_Xog3 · 2022-10-18

**Confidence:** 4
**Correctness:** 3
**Technical Novelty And Significance:** 4
**Empirical Novelty And Significance:** 2
**Recommendation:** 6

**Clarity, Quality, Novelty And Reproducibility:**

The clarity of the paper is generally OK except for some details missing and some notations missing (should be easy to fix).
The quality of the paper is good although it could be improve by adding more baselines and ablations to show better that the method actually improves state-of-the-art. Finally, the paper provides an original contribution and references well previous work.


**Strength And Weaknesses:**

The strengths of the paper are:
* tackle a very relevant and impactful problem
* provides a novel technical contributions with several ideas that could be leveraged in future work (refinement with gradient-boosting, use of EI equivalent likelihood-free criterion)
* good coverage of experiments: a reasonable range of benchmarks/different blackboxes are considered

Weaknesses:
* lack of runtime analysis: no runtime is given for how long the method takes to return suggestion (the appendix mentions 2048 epochs on top of that, a gradient boosting tree is fitted). However, this can significantly worsen the results if they were reported against wallclock time (as proposed in other works) since then almost all time will be spent in fitting models and the method would likely underperform random-search. Without additional details, one cannot assess if the method will have any practical relevance (a method taking 5 minute to suggest the next candidate will likely not be very applicable).
* the set of baselines is relatively considered is relatively small (only 2 transfer learning and 2 non transfer baselines). They are several baselines that could be easily added: BORE for non transfer, but also BORE/LFBO with search space pruned to bounding box of best previous evaluations [Peronne 2019]. Ideally, another additional transfer-learning baseline on top of this simple bounding-box approach would also be added in order to better assess the quality of the method regarding state-of-the-art (2 methods for transfer is a small number).
* details are lacking, some part of the methods were not clear to me:
  * 4.1: was difficult to get through while it is a key part of the paper, I would suggest adding the dimensions for the different variables and give the final expression of the classifier that sums its two input (which is given in 4.2). In any case, the text alone was not sufficient for me to get exactly how $\phi \dot z_t$ is obtained as I could see several ways to achieve this depending on input dimensions.
  * 4.2: the expression of L^LFBO given in Eq. (3) does not take arguments as input
  * 4.3: how exactly is the Gradient boosting combined with the classifier was not entirely clear to me. I would recommend to write it down formally rather than with words (given the current text description, there could be many ways on how a GB would be fitted to reduce the residuals)
* the set of ablations is small and several complexity are not justified (for instance the approximation done instead of the direct optimization of (3), see additional details for a bigger description on this.

**Summary Of The Paper:**

The paper proposes a new approach to perform transfer-learning for Hyperparameter optimization where offline evaluations are used to fasten the tuning on a new task. The method proposes to leverage recent work that uses classification to train surrogates by learning to classify good/bad configuration. In particular in this work, the classifier takes a combination of global (task agnostic) and local (task specific) features and is learned with a Bayesian Logistic Regression layer which is made possible with the use of several approximations.
In addition, a gradient boosting approach is used to improve the final fit of the model at each iterations. Experiments are conducted on several tabular benchmarking suites  (hpobench, mlbench) and also artificial examples to study the effect of noise levels. The method proposed is show to be competitive or better than the baselines proposed.

**Summary Of The Review:**

The paper provides an interesting and novel method to a relevant problem with several ingredients that may be leveraged by future work (for instance using GB to improve surrogate quality). However, as it stands the experimental section has only too little baselines to really its performance against state-of-the-art (only 2 baselines for transfer) and some part of the models are unclear. I believe those could be potentially addressed before camera ready.

Additional details:
* many typos, it would be valuable to run a spell-check:
  * meta-leared
  * observaitons
  * This benchmarks
  * Coppola
  * ensmeble
* Eq (3): interestingly, the scale is back in the loss (but the surrogate dont have to predict values proportional to it)
* A potential ablation would be to use BORE to illustrate the benefit of using LFBO in your case, other possible ablations I was wondering about were: using only local/global part of the model, fitting (3) directly as mentioned in the begining of 4.2. The latter would be important to have given that otherwise all the complexity of the approximations (laplace and logits) would not be justified.

---

> ### Author Response · Authors · 2022-11-19
> **Response to Xog3**
>
> We want to thank you for your positive comments and constructive feedback. We will now try to address your questions and concerns.  We would also like to highlight the general response to all reviewers.
>
> 1. runtime analysis
>
> We added several plots to the appendix that show the median time of all of the algorithms to propose the next candidate for two of our benchmarks (they look similar on the other ones that is why we do not want to add all plots). We also show the regret as a function of the estimated wallclock time (estimated because the surrogate benchmarks return an estimated cost for the training) to present the results in a slightly different angle. As the results shows, our methods adds a small overhead to BORE and LFBO, but is considerably faster than other meta-learning models.
>
> 2. limited baselines
>
> We agree that the number of baselines in our initial submission was limited to the most similar methods and needs to be extended. To that end, we added 2 more non-meta-learning methods and RGPE (Feurer et al.) as another meta-learning baseline. The combination of some search-space pruning methods like the one by Perrone et al. with BORE or LFBO is a very good suggestion and we would strive to add this to the camera ready copy (if the paper gets accepted), but we were not able to add this during the rebuttal period as there is no open source implementation of the method that we could easily use in our experimental pipeline. But we hope that with the inclusion of RGPE, we now include the most relevant meta-learning methods for BO where the surrogate model is informed by historical data.
>
> 3. clarity in section 4
>
> We hope we were able to improve the clarity in section 4. Some of the points raised in the review will also be answered by the code that we will release upon acceptance of the paper.
>
> 4. unjustified complexity and missing ablation study
>
> By adding all combinations of Gradient Boosting/No Gradient Boosting and Thompson Sampling/No Thompson Sampling to the plot and also showing on one simple benchmark what happens if the meta-model is not trained, we hope that we can help readers to understand the contribution of each component. We have good reasons for each of our design decisions and hope that we were able to express them more directly in the latest revision of the paper.
>
> Most importantly, let's briefly discuss the two main sources of complexity:
> a) The Bayesian approximation to the distribution of the weights z is not to just add complexity, but we have seen in early experiments that fitting the model via equation (3) directly without any uncertainty and regularisation, leads to an over confident model in early iterations and poor performance.
> b) The introduction of Thompson Sampling was motivated by the same issue, as we saw that the prior can sometimes be too strong to overcome an initially sampled points, which again can result in poor performance, especially in the regime of little meta-data.

---

> ### Comment · Reviewer_Xog3 · 2022-11-21
> **Acknowledging the author answer.**
>
> I acknowledge the authors answers.
>
> My points/concerns about the runtime analysis and ablations were addressed with their answer and additional experiments.
>
> The authors added one transfer method after the rebuttal (in addition to 2 non transfer baselines but those are less relevant/performant in the presence of offline evaluations). However, my concern about lack of baselines remains as only few transfer methods are considered. I will encourage the author to add bounding-box baseline as running methods such as BORE on a bounding box of best previous evaluations is straightforward (also public implementations are available, see for instance this one https://github.com/awslabs/syne-tune/blob/main/syne_tune/optimizer/schedulers/transfer_learning/bounding_box.py).
>
> While not all my points have been addressed, the authors clarified some aspects and I updated my score accordingly.

---

> > ### Author Response · Authors · 2022-11-25
> > **Response to Xog3**
> >
> > Thank you for your response and making us aware of the opensource implementation of the search space pruning methods by Perrone et al. (2019).
> >
> > We combined it with the gradient boosting variant of LFBO and ran it on all the benchmarks. We would call this LFBO with bounding box search space pruning variant as LFBO-BB for convenience. Unfortunately, we cannot update the paper, but the results can be summarized as follows:
> >
> > 1. In the HPO benchmark, the LFBO-BB have similar performance as the RGPE baseline except for the protein problem, where the end performance is closer to random search despite the fast convergence in the first 50 steps.
> >
> > 2. For the NASBench201 experiments, the LFBO-BB method constantly achieve the best meta-learning performance and converges quickly in the first 10 iterations. However, after that the performance plateau and doesn't change until the end of the optimization. Its end performance is slightly worst than the normal LFBO method.
> >
> > 3. In the MLBench problems, the performance of LFBO-BB is close to GC3P across almost all the problems except for the logistic regression problem, where its performance is closer to random search.
> >
> > 4. As for the synthetic functions benchmark, the LFBO-BB variant performs similar to normal LFBO except for the Hartmann3 function, where it has performance closer to GC3P.
> >
> > We conclude from this, that search space pruning works very well for a large number of related-tasks with good coverage on each, but falls short on problems with less data and not localized optima. Especially, the simple search space pruning method tends to ignore the region of interest for the target problem, when the meta-data doses not have a good coverage of the task distribution, leading to saturated performance early on. In general, our method still remains robust and favorable across all benchmarks, although the search space pruning matches its performance on some benchmarks.

---

### Official Review · Reviewer_Xo11 · 2022-10-21

**Confidence:** 3
**Correctness:** 3
**Technical Novelty And Significance:** 2
**Empirical Novelty And Significance:** 2
**Recommendation:** 5

**Clarity, Quality, Novelty And Reproducibility:**

Clarity: The paper is generally hard to follow, many concepts require prior knowledge and are not sufficiently explained in the paper. More detail below.

Novelty: The novelty aspect is low as a similar idea has been explored by Volpp et al. (2020). The proposed meta-learning framework is almost a direct application of Berkenkamp et al. (2021).

Reproducibility: There is no code included with the submission.


**Strength And Weaknesses:**

Strengths:

(+) The problem of (meta) Bayesian optimization is an important problem with several applications.

Weaknesses:

(-) The comparison with existing works is lacking, both in the techniques and in the empirical evaluations.

(-) The lack of a theoretical guarantee (which is common in the BO literature).

(-) The writing can be improved (as detailed below).


**Summary Of The Paper:**

The paper studies the problem of meta Bayesian optimization (BO), which aims to warm-start the BO process by exploiting knowledge from related tasks. In this paper, the authors propose warm-starting the acquisition function, which takes the form of a classifier in the likelihood-free BO setting. Gradient boosting is further incorporated to combat distributional shifts.

**Summary Of The Review:**

The paper presents an interesting idea to tackle an important problem. However, the difficulty or challenges of the problem are not highlighted enough. The comparison with related works is lacking. And the writing can be improved.

Detailed comments:

- The idea of meta-learning an acquisition function has been explored by Volpp et al. (2020). Could the authors elaborate on the differences and advantages of the proposed algorithm to that of Volpp et al. (2020)? In addition, this should also be included as a baseline in the experiment section.

- The paper is generally hard to follow since the readers assume knowledge from several other key papers. I have to read several other works on the BORE framework and BaNNER to understand several parts of the paper. I encourage the authors to make the paper more self-contained by reintroducing concepts from essential related works like Berkenkamp et al. (2021), Tiao et al. (2021), and Song et al. (2022).

- When the input space is a simple Euclidean space without any structure (think of minimizing the function $f(x) = x^2 – 4x$), how does the feature embedding $h(\cdot)$ works? I suppose feature embedding is only useful when there is some structure in the inputs (e.g., images).

- Since BO is a black-box optimization algorithm, can the authors indicate more clearly what are the objective functions that we are optimizing in the experiment section? It is difficult for readers to comprehend and assess results without knowing what are we optimizing.

- From my understanding, UCB is one of the most commonly used acquisition functions, with a nice theory in BO. However, the paper did not mention or compare with UCB. Is there a good reason for that? Can the authors compare the proposed algorithm with GP-based BO methods with EI/PI and UCB?

---

> ### Author Response · Authors · 2022-11-19
> **Response to Xo11**
>
> Thank you for your review and your detailed comments. In the following we will try to answer your questions and address your concerns.  We would also like to highlight the general response to all reviewers.
>
> 1. differences to Volpp et al. (2020)
>
> The idea of meta-learning the acquisition function has been explored by Wistuba et al., (2018), Volpp et al. (2020) and Hsieh et al. (2021).
> Wistuba et al., 2018 manually combine the acquisition function based on GP predictions on the sources and the target task while Volpp et al. (2020) and Hsieh et al. directly learn the acquisition function via a neural network based on GP predictions on the target as inputs to the network.
>
> The key difference is that they still operate within the typical GP-based BO framework, while our work directly models the acquisition function without using any surrogate model to approximate the target function as an intermediate step. This has been proven to be equivalent to BO with expected utility function by Tiao et al. (2021), and Song et al. (2022).
> Adding MetaBO by Volpp et al. to our baselines is one of our next steps, because there is an open source implementation. Unfortunately, we did not finish the integration into our experimental pipeline within the rebuttal period. We are committed to adding this baseline until the CRC in case the paper is accepted. After correspondence with some of the authors of MetaBO, we do not expect it to work as robustly and performant across all benchmarks, so the conclusion about our method would likely not be affected.
>
> 2. clarity of the paper
>
> We appreciate the comment of you and the other reviewers about the clarity and the concrete steps towards improving it. We reintroduce and added derivations from related methods, namely TPE, BORE and LFBO in Appendix A. We hope that our latest revision has improved clarity and is generally easier to follow and more self-contained.
>
> 3. features for problems without structure
>
> The features are supposed to form a bases such that logistic regression can classify the meta-data with high accuracy. This means that the structure of the tasks is inferred from previous examples (and not updated on the target/test problem). But the features do not necessarily have follow the same shape as the function ensemble that the problems come from. After all, the model tries to solve a binary classification task rather than a regression problem.
> We have added plots to the appendix that show the learned features for two one-dimensional problems: 1) Forrester with two very likely positions for the global optimum, i.e. a lot of structure and 2) quadratic functions, where the functions share a certain shape, but the optima could be located anywhere in the search domain. In the first case, most features clearly show where the optima typically are and some seem to allow for local adaptation. In the second case, the features still show structures of the problems by showing that the optima could be on either side of the search domain or in a finite domain in the middle. The model is able to infer this structure for the data.
>
> 4. missing clarity on the objectives
>
> We revisited the experimental section and included the actual objectives in the text.
>
> 5. Comparison to GP-UCB
>
> We added a comparison to GP based BO algorithm using Expected Improvement (EI) as the acquisition function to the latest revision. We chose this because BORE, LFBO, and by extension our method are derived from EI rather than UCB. In the BO literature, both EI and UCB are used and often show very similar performances with neither one being strictly better than the other. The comparison with a GP-based method in the experiments highlights how competitive the likelihood-free methods are on the shown benchmarks.

---

> > ### Comment · Reviewer_Xo11 · 2022-11-21
> > **Thank you**
> >
> > Dear Authors,
> >
> > Thank you very much for the detailed response. I acknowledge that the authors partially address my concerns about the comparison with Volpp et al. (2020) and some clarification in the experiments.
> >
> > However, most of my major concerns are not satisfyingly addressed in the response, namely:
> > - Lack of comparison with GP-UCB. GP-UCB is not even mentioned at all in the paper. It won the test-of-time award in ICML and is one of the most (if not the most) influential BO papers, both practically and theoretically. I strongly encourage the authors to compare/survey GP-UCB in the paper.
> > - Lack of comparison with MetaBO by Volpp et al. (2020).
> > - No theoretical regret guarantees.
> >
> > In general, I feel like the 2 major weaknesses that were mentioned still remain:
> > - The literature review can be improved. I feel that existing works are not sufficiently mentioned and compared with, both in the wider field of BO and in the more specific field of meta BO.
> > - Similarly, the experiments can include more baselines such as GP-UCB or MetaBO by Volpp et al. (2020).
> >
> > For the above reasons, I'd like to keep my original score.
> >
> > Warmest regards,
> > Reviewer.

---

> > > ### Author Response · Authors · 2022-11-25
> > > **Response to Xo11**
> > >
> > > Thank you for your acknowledgement and your feedbacks. We would like to answer your questions and address your concerns in the following.
> > >
> > > - Lack of comparison with GP-UCB
> > >
> > > Thank you for pointing out the lack of comaprison with GP-UCB and we added the GP-UCB in the baseline. Unfortunately we can not update the paper, we will try to describe the performance in the following:
> > >
> > > 1. In the HPO benchamrk, the GP-UCB has the performance on par with RGPE but worst than LFBO.
> > >
> > > 2. In the experiments for NASBench201, its performance is close to random search across three problems.
> > >
> > > 3. For the MLBench problems, the GP-UCB has the similar performance as the GP with expected improvement in the paper.
> > >
> > > 4. For the synthetic function benchmarks, GP-UCB attain close results as GP with expected improvement except on the Hartmann3 function, where it performs better than most of the methods except RGPE, yielding a immediate regret around $10^{-5}$ after 64 iterations.
> > >
> > > We conclude from this, in general, the performance of GP-UCB is close to GP with expected improvement in our experiments. Compared to GP-UCB, our method remains favorable for almost all the problems except for some of the the synthetic functions.
> > >
> > > - Lack of comparison with MetaBO by Volpp et al. (2020).
> > >
> > > Due to the time limit in the rebuttal phase, we were not able to incorporate MetaBO into our experiments, but we are currently working on this baseline and will include it in the camera ready copy if the paper get accepted.
> > >
> > > - No theoretical regret guarantees.
> > >
> > > To the best of our knowledge, there are no theoretical guarantees for likelihood-free BO methods like BORE (Tiao et al., 2021) and LFBO (Song et al., 2022), but they show very strong empirical performance in HPO and AutoML benchmarks. The analyzing these methods would require to introduce strong assumptions on the problem and model class, as well as the noise, but one of the strengths of these classification based BO methods is that the do not make strong assumptions, which renders an analytical treatment harder.
> > >
> > > As our method is a likelihood-free BO algorithms enriched with meta-learning, we inherit the lack of regret analysis but also the even stronger empirical performance.

---

### Official Review · Reviewer_YYiE · 2022-10-24

**Confidence:** 3
**Correctness:** 3
**Technical Novelty And Significance:** 2
**Empirical Novelty And Significance:** 2
**Recommendation:** 3

**Clarity, Quality, Novelty And Reproducibility:**

As discussed above, the novelty of this paper is limited since it is based mostly on the two existing works BaNNER and LFBO. I also have several concerns about the proposed solution as elaborated in the above weaknesses. Regarding clarity, as the Laplace approximation is a well-known technique, the paper can reduce the explanation of the Laplace approximation to include more explanation on the regularization (to learn the task representation), the gradient boosting, and the choice of \tau.

**Strength And Weaknesses:**

The main strength of the paper is in the empirical performance which is shown to outperform two of the existing works, ABLR, and GC3P. However, it has several weaknesses as follows.

1. The technical solution is not novel since it is based mostly on the two existing works BaNNER and LFBO.

2. Although the paper claims that the meta-learned classifier can balance between exploration and exploitation, the proposed approach requires a gradient boost method to correct the errors. It means that the meta-learned classifier does not allow exploration properly. This is unlike multi-task GP, where the correlation between tasks can be learned from the data without any additional ad-hoc method (such as gradient boosting) for correction.

3. While Thompson sampling often relies on a good approximation of the posterior distribution, the proposed approach only uses the Laplace method to approximate the posterior distribution which is a very simple approximation method. Better approximation methods such as variants of MCMC and/or variational inference techniques should be applied.

4. The numerical representation of the task (z) is simply optimized to be close to a standard (multivariate) normal distribution. I am wondering if the proposed method can work with a multi-modal task distribution, i.e., prior tasks form 2 clusters where tasks in a cluster are similar and tasks between clusters are different. In a principled Bayesian approach such as multi-task GP, it can correctly correlate a new task with an existing task.

5. It is unclear to me about the choice of \tau (for prior tasks and the current task) in the proposed algorithm.

6. While the paper reviews a lot of related works on meta-learning for BO, the experiments only consist of 2 existing works while ignoring the others such as those using GP surrogates (e.g., running experiments on low-dimensional problems).

7. Section 3.1 lacks many well-known BO methods such as GP-UCB, predictive entropy search, max-value entropy search, and knowledge gradient-based methods.

8. Since BO is about sample efficiency, can gradient boosting work reasonably well with a small training dataset?

**Summary Of The Paper:**

This work proposes a meta-learning method for likelihood-free Bayesian optimization. In particular, the proposed approach is a combination of two existing solutions, a meta-learning approach from BaNNER and a likelihood-free LFBO, together with the gradient boosting method. The proposed solution is able to work with high-dimensional inputs and handle heterogeneous scales and noises across different tasks.


**Summary Of The Review:**

Due to the limited novelty, concerns about the proposed solution, and the lack of baselines in experiments, the paper may require further improvements to fit the conference.

---

> ### Author Response · Authors · 2022-11-19
> **Response to YYiE Part 1**
>
> We want to thank you for your detailed review with many good questions and concerns. We would like respond to those and to present view on these matters.  We would also like to highlight the general response to all reviewers.
>
> 1. low novelty
>
> While our method is clearly based on the existing work of BaNNER and LFBO, it is more than a straight forward combination of the two. We show how to apply the idea of BaNNER to a classifier in a way that requires little to no tuning of the meta-learning, in contrast to the original work, that tuned the model for every benchmark. Also the combination with Thompson Sampling, which can improve the performance for certain benchmarks, is new.
>
> 2. Why does the model require an ad-hoc addition?
>
> You rightfully point out that non-parametric models, like a multi-task GP learn the task correlations directly and do not need additional models for correction when applied to a new task. This comes at the cost of scalability, which was one of our goals. The idea of MALIBO is to condense the large amount of meta-data into a faster, but certainly limited, parametric model. Depending on the amount and the quality of meta-data, this model can quickly reach its limits in terms of predictive accuracy. This is the trade-off for a more scalable solution. To counter this, we propose to add a non-parametric model that fits the target task better if there is a discrepancy between the meta-model and the data. By choosing Gradient Boosting, we picked a fast method that can naturally incorporate the predictions of any classifier as the initial classifier in the ensemble, making it a perfect fit for our method.
>
> 3. Quality of the posterior approximation
>
> We chose the Laplace approximation based on the speed requirements and accepted the presumably slightly lower performance of it compared to more sophisticated methods as VI or MC(MC). Depending on the number of latent features, the results in  "Simple and Principled Uncertainty Estimation with Deterministic Deep Learning via Distance Awareness" by Liu et al. indicate that the Laplace approximation yields very competitive results for these classification problems. We hope that our revised experiments show the utility of Thompson Sampling despite the simple approximation we used.
>
> 4. Performance for multi-modal task distributions?
>
> In the current form, the approach can perform poorly on problems with multiple, distinct task "clusters", as the loss is matching everything onto a unimodal MVN. The paper "Shape your Space: A Gaussian Mixture Regularization Approach to Deterministic Autoencoders" by Seseendran et al. shows how it can be extended to the multi-modal case. For simplicity, we did not include this in the submission, but this extension can be easily integrated into our method. A multi-task GP (MTGP) can, in principle learn this correlation from an arbitrary task distribution, but in practice, the amount of meta-tasks strongly impacts the convergence of a Bayesian Optimization method based on multi-task GPs. A brief discussion about this can be found in "Fast bayesian optimization of machine learning hyperparameters on large datasets" by Klein et al., where the authors conclude that a large number of related tasks converges more slowly due to the many GP hyperparameters that need to be fitted. Additionally, the MTGP would scale poorly with the amount of meta-data.
>
> 5. Choice of $\tau$
>
> The parameter $\tau$ parametrizes the exploration-exploitation trade-off for BORE, LFBO and our method. It relates to the $γ$-th quantile of the observed $y$ values via $\gamma = p(y \leq \tau \vert \mathcal{D}_N)$. For the likelihood-free based methods, i.e. TPE, BORE and LFBO, they first select the quantile $\gamma$ as hyperparameter, and then $\tau$ is then given by $\tau = \Phi ^{-1}(\gamma)$. While some scaling of this parameter throughout the optimization is discussed in "BORE: Bayesian Optimization by Density-Ratio Estimation" by Tiao et al., all three works use a fixed value of $\tau=$. It added the description for how to choose this parameter in the paper and highlighted with a different color. It would be interesting for future work, how this parameter could be varied or should be chosen based on the observations, but we used the same fixed value as prior work for consistency.

---

> ### Author Response · Authors · 2022-11-19
> **Response to YYiE Part 2**
>
> 6. Missing meta-learning baselines
>
> We added RGPE (Feurer et al.) as a suitable GP based meta-learning base line. Given the amount of meta data ( given in the paper in detail for each benchmarks), this seemed like the most appropriate method from the literature that fulfills our scalability requirement for the meta-data, although the trained GP on the test task still scales cubically with the number of observations.
>
> 7. Missing references
>
> Thank you for pointing out that we forgot to reference some important BO-methods, as we were mainly focused on the meta/transfer-learning methods. We have added them accordingly.
>
> 8. Sample-efficiency of GB
>
> Indeed, gradient boosting is known for its scaling potential to large data sets, but the experiments in  "BORE: Bayesian Optimization by Density-Ratio Estimation" by Tiao et al.and  "A General Recipe for Likelihood-free Bayesian Optimization" by Song et al., show that gradient boosting is very competitive in terms of the optimization performance when used in a likelihood-free BO algorithm. Based on the results there, we concluded that gradient boosting is sufficiently robust to serve as the residual error model for us.

---

> ### Comment · Reviewer_YYiE · 2022-11-21
> **Thank the authors for the response**
>
> I would like to thank the authors for clarifying several concerns, e.g., the choice of $\tau$, the sample efficiency of GB in existing works, the need for the GB in correcting the difference between the meta-model and the data, the revised references, and baselines.
>
> However, some of my main concerns still remain.
>
> 1. After reading the response, the paper is still a combination of the BaNNER framework and the likelihood-free BO loss in my view. In particular, the authors do not highlight the specific challenges/novelty of the proposed approach.
>
> 2. The authors claim that the incorporation of Thompson sampling is an advantage. This is, on the other hand, a part that is not convincing to me. As I mentioned in my previous comments, Thompson sampling works when the posterior distribution is well-estimated (for example, in the Thompson sampling work of Kandasamy et al., (2018) that is cited in the paper). On the other hand, this paper approximates the posterior distribution with the crude Laplace approximation (single mode, Gaussian distributed), hence the effectiveness of Thompson sampling is questionable. While I admit that it is computationally expensive to apply MCMC, I believe the VI approach should be more efficient. It is also noted that the related work BaNNER mentioned in this paper uses HMC in its inference.
>
> 3. The author admits that the current approach can only deal with single-modal task distribution. I believe this is quite limited in practice. In fact, this means that the meta-learning model cannot properly encourage exploration (it may be trapped in a mode of a multi-modal task distribution). The extension to multi-modal task distribution (with supporting experiments) is necessary for this work to be practically significant.
>
> 4. My concern about the use of the gradient boosting method remains. Although I agree with the authors that there should be a method to correct the difference between the meta-model and the data, there is still not any further clarification on the use of the GB (the revised section 4.3 remains elusive). In particular, it is unclear how the GB helps trade off between the meta-model and the data. I believe this is an important detail that should be properly described in the paper for implementation.
>
> From the above reasons and the fact that the paper does not provide any source code for reproducibility, I believe the level of detail and the quality of the approach should be improved. Hence, my opinion remains.

---

> > ### Author Response · Authors · 2022-11-25
> > **Response to YYiE**
> >
> > We would like to thank you again for your detailed response and the good questions. We would like to answer your questions and concerns in the following:
> >
> > - Still a combination of the BaNNER framework and the likelihood-free BO loss
> >
> > The method as presented in our submission is surely based on these two methods, but we made significant additions to arrive at a method that performs well across all tasks without additional tuning.
> >
> > 1. We use Bayesian logistic regression to change the regression model to a classification model. While there different ways to do this, we found through our experiments that simple, deterministic classifier lead to overfitting, and not all Bayesian approaches lead to scalable methods.
> >
> > 2. We propose Gradient Boosting as a residual error correction model, which is one of the novel part in the method. We have shown the ablation study of the effectiveness of this residual model.
> >
> > 3. We use Thompson sampling leveraging the Bayesian treatment in Bayesian logistic regression, which can help the model explore the search space efficiently and to parallelize the optimization, as we show in the appendix.
> >
> > - Thompson sampling.
> >
> > We agree that the Thompson sampling requires well-estimated posterior, and the Laplacian approximation is a basic method which only approximates the local structure of the true posterior.
> >
> > However, the meta-training of our model successfully enforces the latent representation of all the meta-tasks to follow a multi-variate normal distribution. Assuming that any new task is similar to the historical ones (which is the main assumption for any meta-learning), the classification problem that needs to be solved during fitting the model satisfies the usual requirements for Bayesian logistic regression, where the Laplace Approximation is a well-established tool.
> >
> > With respect to MCMC sampling as an alternative, Berkenkamp et al. showed with BaNNER, that having a simple Bayesian linear regression layer with learned features was competitive to sophisticated MCMC methods in terms of optimization performance, but orders of magnitudes faster, which was one of our goals. We expect this to carry over to Bayesian logistic regression.
> >
> > Alternative VI approximations would, in principle, be feasible, but we expect them to be less robust w.r.t. their hyperparameters across tasks without offering significantly better performance.
> > For very expensive problems, like the NAS201 benchmark, these more computationally intense alternatives might be applicable, but for broader applications we argue that the simple Laplace approximation is a good compromise between efficiency and accuracy with already good practical performance (as shown throughout our experimental section).
> >
> > - Multi-modal task distribution.
> >
> > As mentioned in the previous answer, it is possible to extend the presented method to the full multi-modal case, by following the work of Saseendran et al. (2022). From our experiments, the results in Figures 1, 11, 14 and 16 are the closest to a multi-modal distribution, where the optima are found in two distinct locations, depending on the task. This is not a true multi-modal problem, as the function ensemble interpolates between these two clusters, but resembles some characteristics thereof. The results show that the non-linear, latent features combined with Bayesian logistic regression and Thompson sampling, are effective at exploring both local optima efficiently. Our method successfully learn meaningful feature representation for modeling both modes of the target function (Fig. 11), and specifically show how the method explores both modes leveraging Thompson sampling in Fig. 14 and 16. We argue that, although there is still room for extension to more accurate multi-modal task distributions,the empirical results show that our method is already practically useful.
> >
> > - Gradient Boosting.
> >
> > Besides revising section 4.3, we also added ablation study in Appendix D.2 where we show that gradient boosting helps the method to optimize when there is a mismatch between meta-data and target function (no meta-data in this case). We will continue to improve the text and also provide open-source implementation for the method once the paper get accepted.

---

### Official Review · Reviewer_YSQy · 2022-10-27

**Confidence:** 4
**Correctness:** 3
**Technical Novelty And Significance:** 3
**Empirical Novelty And Significance:** 2
**Recommendation:** 6

**Clarity, Quality, Novelty And Reproducibility:**

### Questions

* For the experimental results, the variance (or standard deviation) of the experiments (i.e., shaded regions) is somewhat odd. Since $y$-axis is a log scale, the variance at the last of iterations should be larger than the variance at the beginning of iterations.

* A feature extractor can be called as ResNet? According to Appendix C, your feature extractor is different with ResNet. I think that the name should be changed.

* I would like to ask about gradient boosting. Which gradient boosting is used in this paper?

* Following the above question, if you use gradient descent, how did you optimize a function $C$ (Line 10 or Line 12 of Algorithm 1)?

* In the experiments, three proposed method, i.e., MALIBO wo GB, GB, and GB-TS, are tested. Does MALIBO wo GB include TS? Why did not you test MALIBO wo GB w TS or MALIBO wo GB wo TS?

### Minor issues

* In Page 4, $\max(y - \tau)$ seems like a typo; please fix it.

* In the caption of Figure 2, $x_t$ should be $\boldsymbol x_t$.

**Strength And Weaknesses:**

Here I describe the strengths and weaknesses of this paper.

### Strengths

It solves an interesting topic where multiple historical tasks are given, by applying a likelihood-free framework.

The proposed method is quite novel. In particular, a combination of some components such as mean prediction layer, residual prediction layer, and gradient boosting is interesting.

The purpose of the respective components and the respective loss functions is well-described.

### Weaknesses

Presentation and writing can be improved. The current version is okay, but I think it can be polished more.

Iteration budgets for the experiments are too low. I think that you should give a larger iteration budget in order to show the convergence of the algorithm tested.

**Summary Of The Paper:**

The work proposes a likelihood-free Bayesian optimization strategy with a meta-learning scheme. Given multiple tasks, it trains task-agnostic and task-specific components in order to predict a probability that measures how likely a solution is, inspired by BORE and LFBO. By utilizing Bayesian logistic regression, it determines a query point. In addition, the authors use a gradient boosting model to predict a residual of the model. Eventually, the experiments demonstrate the effectiveness of the proposed method, compared to other existing methods.

**Summary Of The Review:**

Please see the above text boxes.

---

> ### Author Response · Authors · 2022-11-19
> **Response to YSQy**
>
> Thank you for your positive feedback and questions. We will try to answer all of them in this reply. We would also like to highlight the general response to all reviewers.
>
> 1. Inconsistent standard deviation?
>
> The standard error in our plot is based on the evaluations and computed for each iteration separately. For a constant standard error, your comment about log-y axis is valid, but most algorithms consistently converge to similar regret values, although the effect of visually larger uncertainties can be seen, for example, in Figure 3a.
>
> 2. Naming of the feature extractor.
>
> Our neural network design follows the one used in the tutorial for "Simple and Principled Uncertainty Estimation with Deterministic Deep Learning via Distance Awareness" by Liu et al., which can be found [here](https://www.tensorflow.org/tutorials/understanding/sngp).
> We change the name of the network to ResFFN-4-64, as the architecture is a Residual Feedfoward Network (ResFFN), with 4 residual feedfoward layers and each has 64 units.
>
> 3. Which gradient boosting is used?
>
> We used the scikit-learn implementation of [GradientBoostingClassifier](https://scikit-learn.org/stable/modules/generated/sklearn.ensemble.GradientBoostingClassifier.html) with the default parameters and used the meta-learned model as the initial classifier. We also added this detail to the paper.
>
> 4. How is the function C (line 10 or 12) optimized?
>
> We follow the approach in LFBO and BORE and optimize the function C by picking the argmax from a large number of random samples, to be precise, we use 5120 samples in our experiments. Using gradient descent is conceivable, but not for the variants that use gradient boosting.
>
> 5. Variants /w and /wo GB and TS.
>
> For completeness, we have added all combinations of Gradient Boosting and Thompson Sampling.

---

> > ### Comment · Reviewer_YSQy · 2022-11-21
> > **Additional questions**
> >
> > Hi Authors,
> >
> > Thank you for your response.  I would like to ask some additional questions.
> >
> > Could you specify how you draw standard deviation?  Did you multiply 1.96 by standard deviation?  I think that there is a way to draw symmetrical error bars in a log scale.
> >
> > Moreover, could I ask why the starting immediate regrets are different across algorithms?  If you fixed initial points, they should be identical.

---

> > > ### Author Response · Authors · 2022-11-25
> > > **Response to YSQy**
> > >
> > > Thank you for your additional questions. We will try to answer them in this reply.
> > >
> > > - Could you specify how you draw standard deviation?
> > >
> > > In our plots, we actually show the mean and standard error from our experiment results, instead of standard deviation. We clarify that in third paragraph at Sec 5.
> > > On a linear scale, this leads to symmetric uncertainty regions that are distorted on a logarithmic scale.
> > >
> > > - Why the starting immediate regrets are different across algorithms? If you fixed initial points, they should be identical.
> > >
> > > The reason for this is that we didn't fix the initial points of any algorithm. Most of the algorithms, including the GP-based methods and the non-meta-learning likelihood-free methods, first draw random samples in the first few iterations before optimizing. In contrast, our GC3P and our method improve over random samples by informed guesses. For our method, the task-agnostic mean prediction layer can propose a promising configuration at the first step already. Therefore, the starting immediate regrets vary across algorithms.

---

> > > > ### Comment · Reviewer_YSQy · 2022-12-01
> > > > **Thank you for your response**
> > > >
> > > > Hi Authors,
> > > >
> > > > Thanks for your response.
> > > >
> > > > I acknowledge that I have read your response and the other reviews.

---

### Author Response · Authors · 2022-11-19
**General response to all reviewers**

We want to thank all reviewers for their constructive feedback and critical questions. We were pleased to find that the field of our work is recognized as important and hope to show that our contribution should be considered for publication at ICLR with our revised submission. In particular, we have

 - improved the clarity and writing throughout the whole paper, with a focus on the presentation of our contribution and the discussion of the experimental results.
- added BORE and a standard GP based BO variants as non-meta-learning baselines and RGPE as another GP-based meta-learning method to all the experiments.
- added results that show the optimization progress as a function of the simulated wallclock-time making the optimizer overhead explicit and showing the scalability of our method (see Appendix C).
- added visualizations of the learned features and the first iterations of the optimization for two 1-dimensional problems to illustrate the type of features learned and how the algorithms suggests new points (see Appendix D and E).
- added an ablation study to investigate the contribution of the individual components and how they affect the performance (see Appendix D.2).
- added the more details and derivations for other related methods, namely TPE, BORE and LFBO. (See Apppendix A)

We will answer every reviewers questions and address their individual concerns in separate replies to the respective reviews.

---

### Decision · Program_Chairs · 2023-01-20

**Decision:**

Reject

**Justification For Why Not Higher Score:**

See reviews. The main issue is lack of novelty, given previous work. While there is one new idea (residual correction by way of xgboost), this is too little to justify acceptance.

**Justification For Why Not Lower Score:**

N/A

**Metareview: Summary, Strengths And Weaknesses:**

This paper is proposing a Bayesian optimization method, where the acquisition function is given via a classifier separating good and bad evaluations (observations split along a percentile), an idea introduced in TPE, generalized in BORE and other work. It makes use of transfer from previous tasks, again close to earlier related work.

As reviewers point out, there are some novel ideas here, for example the separation into task agnostic and specific parts to the surrogate, and the use of xgboost as a non-parametric representation of residuals. However, given previous work, these novelties are not enough to justify acceptance. The empirical evaluation is also somewhat limited, missing a number of baselines.

Short of a novelty that tackles a major limitation of prior work (e.g., in the high-dim BO context, or conditional spaces, like the original TPE tried to do), this work could become stronger by tackling harder real-world problems.